# A study on the evolution of original sites of fortifications from the perspective of Historic Urban Landscape: Cases of Paris, Beijing, and Moscow

Mo Xu *, Haolin Zhu

School of Sciences for the Human Habitat, University of Chinese Academy of Sciences (UCAS), Beijing, China

* 15801632535@163.com

## Abstract

The Historic Urban Landscape theory underscores the importance of historical stratification processes in shaping the overall value of urban heritage. Building on this perspective, this study examines the stratification and evolution of the original sites of fortifications in Paris, Beijing, and Moscow, three global cities. First, historical research is conducted to identify the key moments of significant changes and their contextual backgrounds in these cities' original sites of fortifications. Second, quantitative analysis is applied to calculate the changes in functional proportions across different periods. The findings reveal that, although the timing of major transformations varies among the three cities, all have undergone three distinct stages: demolition and planning, development and construction, and reflection and renewal. This evolutionary process is closely tied to the urbanization trajectories of each city, with the driving forces exhibiting notable commonalities and patterns.

## Introduction

The original sites of fortifications, areas of land left by the demolition or natural erosion of fortifications during urban development, mark the precise locations of past fortifications and associated defense structures. These critical historical components not only safeguarded cities, but also embodied their history, culture, and social structure. However, defortification waves of the 19th and 20th centuries led to their dismantling due to political, military, and economic factors. Today, redevelopment and large-scale modernization have erased much of their historical significance. They now face significant environmental pollution, traffic congestion, and pedestrian inaccessibility, making the original fortification sites nearly uninhabitable and were perceived as negative urban areas.

Introduced in 2011 by UNESCO, the concept of Historic Urban Landscape (HUL) offers a new perspective on original fortification sites. HUL is defined as "the urban area understood as the result of a historic layering of cultural and natural values and attributes, extending beyond the notion of 'historic center' or 'ensemble' to include the broader urban context and its geographical setting" [1]. It is not a static and isolated entity but a dynamic and living

**Data availability statement:** All relevant data are within the paper and its Supporting Information files.

**Funding:** The author(s) received no specific funding for this work.

**Competing interests:** The authors have declared that no competing interests exist.

organism encompassing both material and immaterial elements that have accumulated within the area.

The historic urban landscape is not only a heritage but also a research methodology that emphasizes the continuous process of urban development and acknowledges the stratification of urban heritage. These historical layers collectively constitute the overall value of urban heritage, with their diversity and profound significance reflecting the cultural accumulation and transformations of different historical periods. Research on historical stratification serves as a crucial approach to uncovering and systematically assessing the value of urban heritage [2]. It provides an objective perspective on the various phases of urban evolution, respecting the cultural significance of distinct historical contexts and avoiding omissions or biases in value judgment. Moreover, the patterns of transformation embedded within historical stratifications not only reveal the logic of past urban development but also offer essential insights into the potential trajectories of future urban change.

Today, although most fortifications have vanished, their original sites maintain linear spatial form and some historical relics. Furthermore, the demolition of these fortifications and the subsequent planning and development of their sites have birthed many modern urban planning ideas and architectural legacies, reflecting shifts in urban economics and public ideologies. Moreover, from urban perspectives, the original sites of fortification now thrive as bustling urban centers, attracting substantial human traffic and social activities. In conclusion, original fortification sites hold considerable historical, social, and economic value, constituting a rich and diverse HUL. However, urgent research into their evolutionary processes and prospective development trends is essential for protecting and revitalizing the region [3].

In cultural heritage, fortifications are frequently discussed; however, their original sites are seldom explored independently. Instead, they are integrated into broader fields, such as urban morphology, urban geography, urban planning, and cultural heritage. Conzen's "Inner Fringe Belt" [4] concept in urban morphology encompasses these original sites. Mladen's research indicates that urban green spaces often develop on or around the original fortification sites, forming a crucial component of the Inner Fringe Belt [5].

Few studies have addressed original fortification sites for cross-city comparative analysis, but more research exists on individual cities. Jean-Louis Cohen et al. analyzed Paris's fortification sites' evolution over a century and a half, discussing their impact on Paris's urban space, architecture, and residents' lives [6]. TOMATO Architects studied the history and development of Paris's "Boulevard Peripheries," proposing strategies for city boundary optimization and overall urban development [7]. Tanja Winkler detailed the formation process of Vienna's fortification sites, now Vienna's Ringstrasse, and its socio-political and economic influences on urban planning and architecture [8]. Lei Li analyzed the development and evolution of Beijing's Ming and Qing fortifications, now the Second Ring Road, and introduced urban ring greenway concept, an ideal model for the Second Ring Road's landscape construction [9]. P.V. Kopylov examined the "My Street" renovation project's impact on Moscow's fortification sites, now the Garden Ring, revealing issues with uneven greening of the Garden Ring [10].

In conclusion, current research on original fortification sites primarily focuses on case studies of specific cities, overlooking their universal development traits and future transformation directions as special urban areas. Therefore, considering the similar development patterns and consistent challenges faced by Original Sites of Fortifications in most cities, this paper will employ the Historic Urban Landscape (HUL) approach to examine the similarities and differences in their development processes in three major global cities. The study will focus on the demolition of fortifications, planning and construction, as well as subsequent renewal and transformation. The objective is to provide valuable insights for the construction and renovation of Original Sites of Fortifications in other cities.

## Methodology

The original fortification sites in Moscow, Paris, and Beijing were chosen for comparative analysis for three key reasons. First, these cities dismantled their fortifications and constructed ring roads, triggering numerous urban issues and prompting reconsideration of site usage, leading to renewal and practical applications. Second, as major world cities and national capitals, the land use of these sites reflects their countries' social and historical backgrounds, as well as their developmental trajectories. Lastly, all three cities prioritize preserving their historical and cultural heritage and possess significant international influence. Their strategies to protecting and updating these sites offer valuable lessons and reflections to other cities.

Due to the Historic Urban Landscape theory emphasizing the stratification in the process of urban development, this study conducts a comparative analysis of the evolution of the original sites of fortifications in Moscow, Paris, and Beijing. The research adopts a mixed-methods approach combining qualitative and quantitative analysis: the qualitative analysis reviews the historical evolution of the original sites of fortifications in the three cities, exploring the key events and causes that triggered significant changes; the quantitative analysis uses the Open Street Map platform and historical maps of the three cities from the 19th to the 20th century to create vector data of the original sites of fortifications in different periods using ArcGIS Pro and calculate the functional proportions for each period. This study aims to uncover the patterns and driving forces behind the evolution of the original sites of fortifications in these cities, providing valuable insights for the protection and planning of similar sites in other cities in the future.

## The evolution of the original sites of fortifications in three cities

### The evolution of the original sites of fortifications in Moscow

**The demolition of the Zemlyanoy Rampart and the birth of the garden ring.** The Zemlyanoy Rampart, initially a wooden fortification known as Skorodom, was the final defensive fortification protecting Moscow. Constructed by Tsar Fyodor Ioannovich in 1591, it faced destruction by Polish forces in 1611 [11]. To bolster the city's defenses, Alexei Mikhailovich oversaw its transformation into the Zemlyanoy Rampart from 1638 to 1641. Spanning 15 km, it boasted 57 towers, 12 gates, 9 bastions, and a moat (Fig 1a). However, with the advent of military reforms, the Zemlyanoy Rampart was rendered obsolete and repurposed as the city's customs boundary. By the late 18th century, the fortification had progressively crumbled, with some parts demolished to create spacious squares and thoroughfares [12].

However, during the French occupation of Moscow in 1812, fire destroyed much of the area within the city walls [13]. In post-war reconstruction efforts, the Moscow Building Committee recommended demolishing the walls, filling the ditches, and constructing a broad circular street in their place. Previously, houses stood 60-70 meters apart on either side of the wall; now the plan allocated the central 25 meters to city roads and sidewalks. Homeowners could then arrange front gardens according to their preferences in the remaining space. By 1830, most of these gardens were completed, marking the birth of the prototype of the "Garden Ring" (Fig 1b) [14].

### The disappearance of gardens and the transformation of ring roads

At the beginning of the 20th century, tram line "B" replaced horse-drawn trams (konka) on the Garden Ring, while multistory buildings for commercial, administrative, and residential purposes began to be constructed on both sides of the ring, replacing older

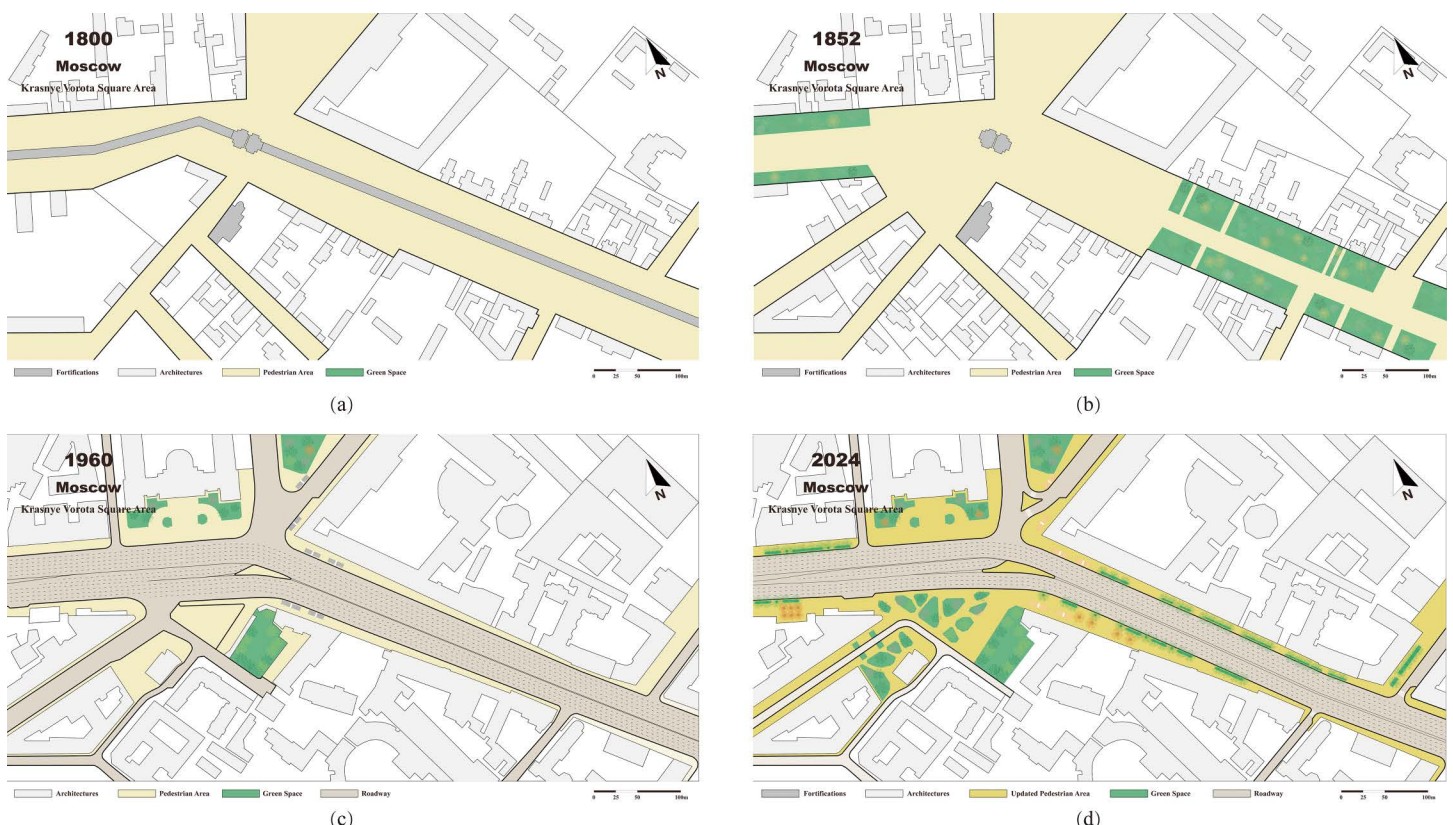

**Fig 1. Evolution of the Original Sites of Fortifications in Krasnye Vorota Square, Moscow:** (a) Plan of Krasnye Vorota Square in 1800, before the demolition of the fortifications; (b) Plan of Krasnye Vorota Square in 1852, where the original sites of fortifications were converted into the Garden Ring Road with gardens; (c) Plan of Krasnye Vorota Square in 1960, when the Garden Ring Road was widened, and the former gardens disappeared; (d) Plan of Krasnye Vorota Square in 2024, where the lanes of the Garden Ring Road were narrowed, pedestrian walkways were expanded, and gardens increased.

low-rise buildings. The 1935 General Plan for Moscow proposed the widening of Garden Ring by removing its gardens, demolishing significant buildings on the ring to create broad squares, eliminating trams, and introducing Stalinist architecture. In 1937, the Garden Ring had been widened, its gardens removed entirely, and new bridges over the Moscow River [15].

After World War II, extensive urban development began and around the garden rings. Between 1948 and 1954, three high-rise Stalinist buildings were constructed on the ring: the Foreign Ministry of Smolenskaya Square, a residential building in Kudrinskaya Square, and a residential and administrative building in Krasnaya Vorota Square. To improve the public transport system, construction of a southern section of the metro circle beneath Garden Ring started in the early 1950s. Additionally, to address increasing traffic volumes, the government constructed tunnels, overpasses, and underground passages at major intersections along the ring during the 1950s and the 1960s (Fig 1c).

Owing to the Garden Ring's excessive number of lanes, some sections have up to eight lanes, encroaching on the city's public space. This has resulted in severe shortage of sidewalks and green spaces at the original fortification sites, exacerbating urban noise and air pollution. Additionally, despite the presence of underpasses, the cost for pedestrians to cross the ring greatly rises, creating a substantial division in the urban space on either side.

### The transformation strategies for the original sites of fortifications in Moscow

Car-centric planning principles turned Moscow's Garden Ring into a high-traffic thoroughfare, sacrificing pedestrian space and eliminating the original gardens, diminishing its appeal and sustainability. Therefore, to improve the street environment, in 2014, the Moscow government launched a major urban renewal project, "My Street," inspired by Jan Gehl's "people-first" urban design philosophy [16]. The project aims to reintroduce gardens to the Garden Ring, enhancing the walking experience and restructuring transportation nodes.

Firstly, to optimize traffic design. the "My Street" project enhanced Garden Ring's walking experience by standardizing the lane width and number and increasing green space. Previously, the lanes were 3.5 meters wide for regular lanes and 3.7 meters wide for bus lanes, with varying lane counts between five and eight. After the renovation, regular lanes were reduced to 3.25 meters, bus lanes to 3.65 meters, lanes per direction to five, and sidewalk parking eliminated. This created more room for sidewalks and bike lanes Simultaneously, a large number of trees were planted on both sides of the Garden Ring, forming a natural ecological barrier that enhanced the safety and comfort of the pedestrian areas. Additionally, permeable paving technology mitigated the heat island effect, and additional intersections were added to improve the convenience of crossing the ring (Fig 2).

**BEFORE**

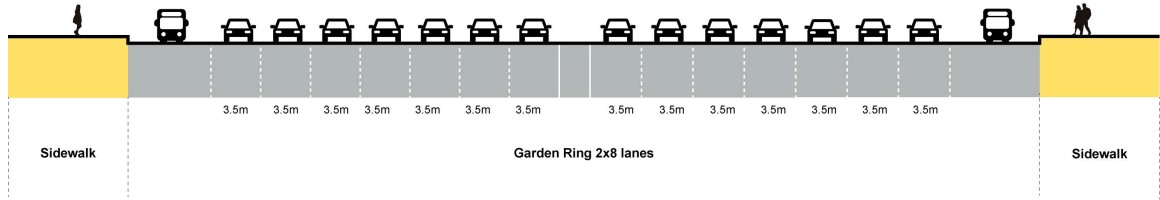

**AFTER**

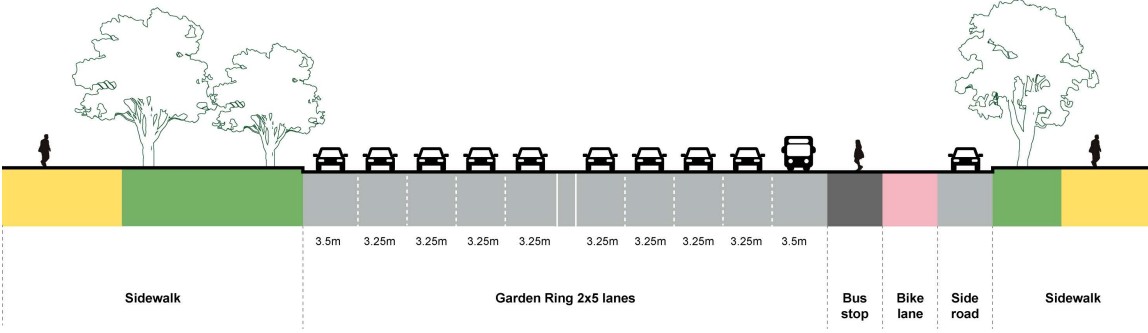

**Fig 2. Diagram of the Gargon Ring Transformation in Moscow.**

In addition, The Garden Ring was enhanced by creating gardens, recreational areas, sports fields, and exhibition spaces to upgrade the public space nodes. First, notable improvements were made around metro stations, such as the transformation of the chaotic parking lot near Krasnye Vorota metro station into a lush urban public space. Dobryninskaya Metro Station's vicinity was reimagined as exhibition venue. Additionally, attention has been paid to the fragmented spaces around the ring road: Unused land in the central part of Zhitnaya Street has been transformed into a quiet urban garden, and an area under the Krymskaya overpass near the park Kultury metro station was planned as a skate park (Fig 1d).

## The evolution of the original sites of fortifications in Paris

### The demolition of the thiers wall and the green belt formation

In the 1830s, owing to tensions between England and France over Egypt, Louis-Philippe ordered the construction of the Thiers Wall from 1841 to 1844. This 10-meter-high, 35-km-long wall included 52 gates, 94 bastions, a 15-meter-wide moat, and glacis. Additionally, a military road was built on the city side, later becoming Boulevards of the Marshals in 1860 [17]. On the suburban side, a 250-meter-wide non-building zone was established [18]. However, the Franco-Prussian War of 1870 proved the fortifications' futility. Additionally, as Paris's population increased, the non-building zone became occupied by slums, hindering urban expansion, sanitation, and recreational activities (Fig 3a).

Since 1880, Paris has seen ongoing discussions regarding the demolition of fortifications and urban planning. In 1908, politician Louis Dausset proposed a "green belt city" plan, laying the groundwork for today's "green belts" and influencing the 1919 legislation [19]. Considering that parts of the original fortification sites had been occupied by railways, cemeteries, and schools, the government organized another urban planning competition in 1919. In 1924, Louis Bonnier's planning project was approved, which systematically graded the road network and housing projects, and designed two parallel urban bands: one consisting of social housing (HBM) and its associated social service facilities; the other comprised a continuous ring of green spaces, including sports fields, squares, parks, and walking paths (Fig 3b) [20].

### The construction of two parallel Urban strips and a boulevard périphérique

From 1926 to 1939, Louis Bonnier's plan led to the construction of a substantial number of low-cost housing units (HBM) on the inner urban strips of the original fortification sites. These included ordinary low-cost housing for workers and employees (HBMO), middle-income housing for the moderately wealthy (ILM), and improved low-cost housing (HBMA) positioned between the two. Typically, ILMs are located near city gates and large forests, whereas HBMs are built on lower-value lands. However, some developers prioritized profits over quality, resulting in HBMs that did not meet health standards and were inadequately equipped [21]. On the outer green belt, 135 hectares of land were developed into parks, racetracks, sports stadiums, and other public facilities through expropriation, although much of it remained undeveloped because tenants did not relocate. On October 11, 1940, the Vichy government enacted a law concerning the original fortification sites, allowing the expropriation of land requiring sanitation and authorizing the use of force. Thus, during World War II, residents of the remaining 259 hectares were evicted under the pretext of "anti-hygiene" [22].

Due to a severe housing crisis after World War II, on February 7, 1953, the Paris city government passed the "Lafay Law," allowing residential construction on 20% of the land in the original "non-building zone" [23]. In 1954, the City Council approved a plan to build approximately 4,000 housing units in seven areas of the Green Belt, leading to the occupation of the original green spaces. Simultaneously, in 1954, to address growing traffic pressures,

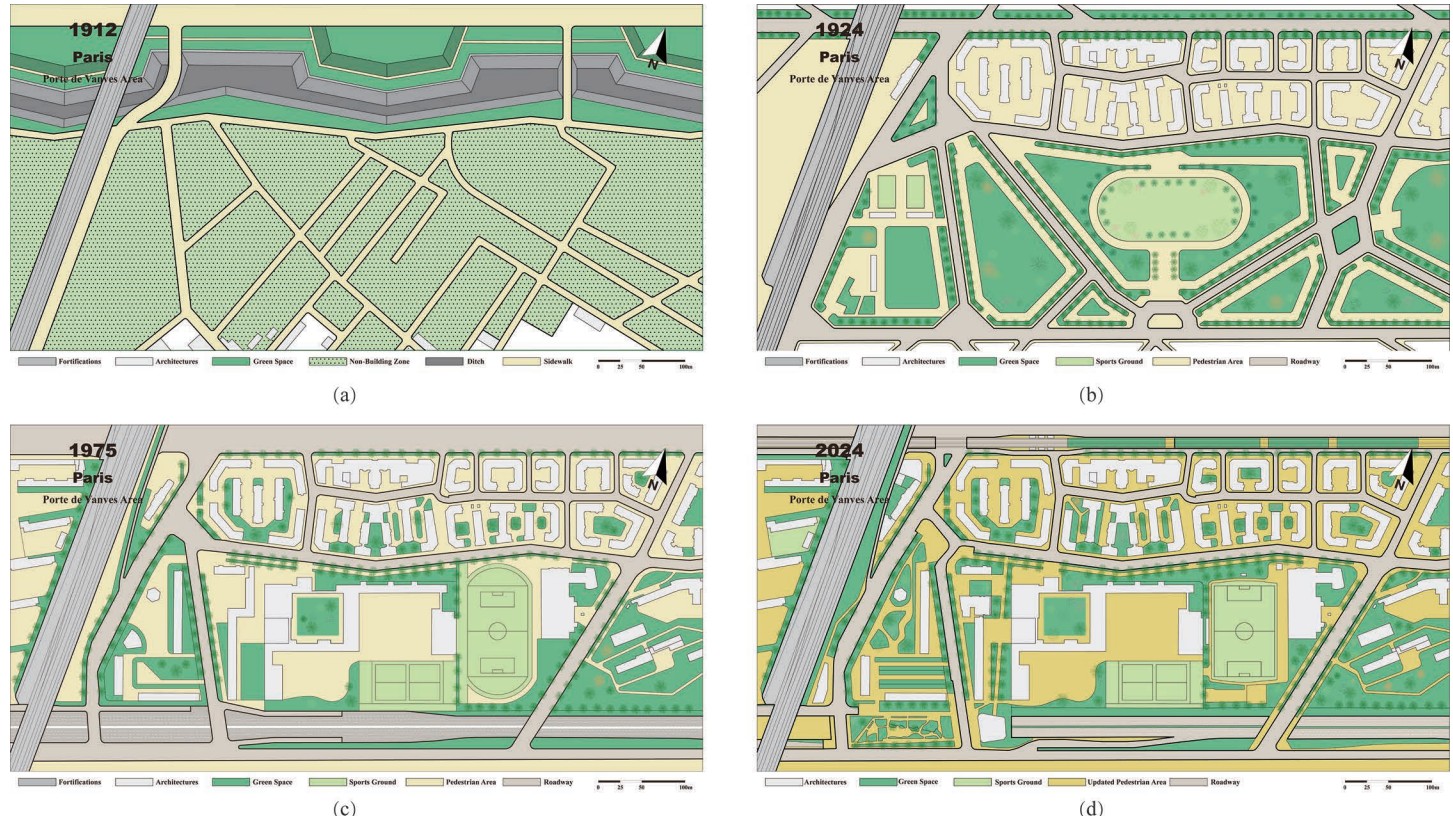

**Fig 3. Evolution of the Original Sites of Fortifications in the Porte de Vanves Area, Paris:** (a) Plan of the Porte de Vanves area in 1912, before the demolition of the fortifications; (b) Urban planning map of the Porte de Vanves area in 1924, designed by Louis Bonnier; (c) Plan of the Porte de Vanves area in 1975, where part of the planned green belt was occupied by the Boulevard Périphérique, as well as residential and public buildings; (d) Plan of the Porte de Vanves area in 2024, where parts of the Boulevard Périphérique have been covered and transformed into public spaces.

the government decided to construct a peripheral expressway. The expressway, implemented from 1956 to 1973, had a road width of 35 to 60 meters, further reducing the size of the greenbelt [24]. It included at-grade, elevated, and tunnel sections. Initially intended to protect adjacent social housing from pollution, the expressway proved minimally effective and created a significant divide within the city that was difficult to cross (Fig 3c).

## The transformation strategies for the original sites of fortifications in Paris

To address environmental pollution and spatial fragmentation, Paris City government abolished the Lafay Law in 1985 and established a 14-kilometer noise barrier to reduce pollution. In 1988, the Paris Urbanism Agency (APUR) proposed "The Objective Framework for the Development of the Crown in Paris," guiding the land-use functions of the original fortification sites for the first time. Three projects covering the city gates were included in the Grand Urban Renewal Project (GPUR) for 2000-2006 [25]. The establishment of the tram line (T3) in 2006 also fundamentally improved the ring area's accessibility. In recent years, renewal objectives have refocused on the early "Green Belt City" concept. In 2013, APUR launched an initiative titled "Revival of the Green Belt," aimed at enhancing biodiversity, functionality, and accessibility in the Crown, thereby rebuilding connections between central Paris and Grand Paris [26]. The 2024 Paris Olympics and Paralympics provide new opportunities for the

development of the Crown. APUR has explored the possibilities of constructing pathways in the green belt that integrate sports, art, and ecology in Green Belt Active Route: Project Plan 2024," and the post-Olympic transformation direction of the Boulevard Périphérique in the "White Paper - New Green Belt and Transformations of the Boulevard Périphérique [27]. In summary, the update strategies for the original sites of fortifications in Paris during this stage focused on transforming the ring road, enhancing regional connectivity, and enriching urban functions.

Firstly, by reducing the number of lanes and limiting the speed on the Boulevard Périphérique, pollution is reduced, green travel is promoted, and more green spaces are available. According to the recommendations in the "White Paper - New Green Belt and Transformations of the Boulevard Périphérique" published by APUR in 2022: After the Olympics, the four lanes of the Boulevard Périphérique will be reduced to three, with the left lane uniformly transformed for carpooling, buses, and taxis to improve transportation efficiency, alongside reducing the speed limit to 50 km/h. Future plans may include creating intersections, traffic lights, and pedestrian crosswalks (Fig 4)

**BEFORE**

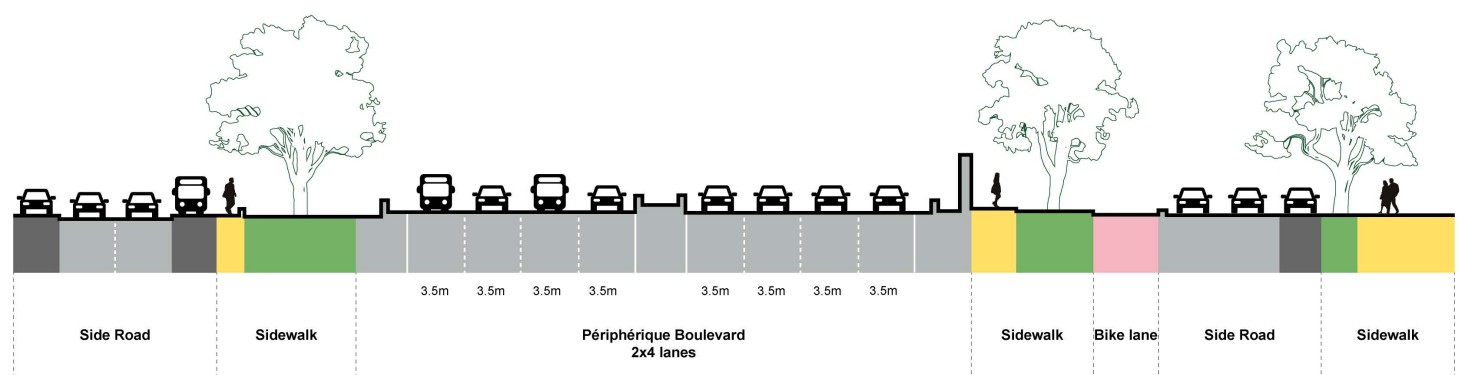

**AFTER**

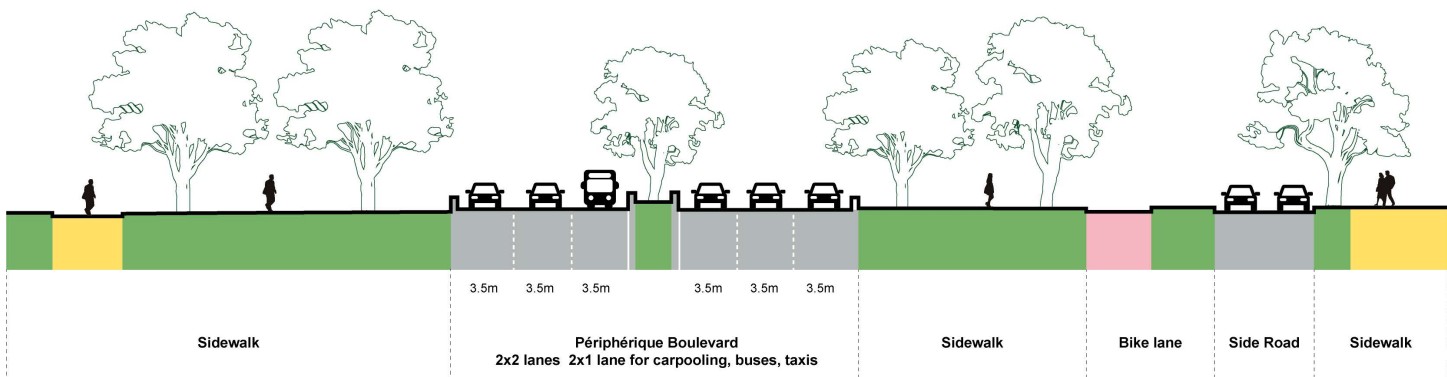

**Fig 4. Diagram of the Paris Boulevard Périphérique Transformation.**

Secondly, enhancing the connectivity of urban areas on both sides of the ring road is essential. Since 1988, Paris has sought to address the challenge of reconnecting urban spaces divided by the ring road. In 2002, GPUR implemented a strategy to bridge the gap by covering trench-like roads in areas such as Porte des Lilas and Porte de Vanves with platforms, creating open spaces or buildings for community use (Fig 5b and c). Currently, 22 gate areas have either been transformed or are planned for renovation. For example, in Porte de Montreuil, a large green square will cover the area. At Porte de Brancion, the "Woodeum" project above the Boulevard Périphérique is a low-carbon solid wood residential building for young people.

However, due to technical and economic constraints, a universal approach of full coverage is not always feasible. Instead, some gate areas have adopted alternative strategies to optimize existing conditions by enhancing green landscapes and creating ecological corridors (Fig 5e). For instance, at Porte de Maillot, the removal of the roundabout restored the city's historical axis, reducing traffic areas by 41% and significantly increasing green space, which now extends to the Bois de Boulogne.Another major strategy involves creating new crossing pathways, including pedestrian bridges and the utilization of spaces beneath overpasses to establish public areas (Fig 5a and d). For example, the Passerelle Claude Bernard in northeastern Paris spans the ring road, connecting Paris with the Aubervilliers district, while the Passerelle du Cambodge in the south links Paris' Cité Universitaire with Gentilly. In Porte Pouchet, the area beneath the overpass has been transformed into a large public space featuring plazas, green areas, and sports facilities, encouraging pedestrian movement and integration across the ring road.

Finally, by enriching, integrating, and enhancing urban services in the ring road areas, the planning concept of a "15-minute city" is realized (Fig 5f). First, the Olympics enhanced the sporting function of the Crown. The government not only constructed new gymnasiums but also added various sports and entertainment facilities. Relying on existing parks, sports facilities, streets, and other cultural and leisure venues, a new sports corridor was created to provide an ideal sports venue for the surrounding communities. For example, in the Porte de la Chapelle area, a multi-story parking lot was demolished to construct a sports arena for the 2024 Paris Olympics. After the event, the venue will be repurposed as a basketball court and concert hall, while also incorporating retail and dining facilities. Additionally, the surrounding roads have been restructured and integrated, creating significant public space. This reconfiguration of building functions has revitalized the area, breathing new life into the district. Additionally, logistics services are one of the future development directions for the Crown area and are intended to replace abandoned railway sites, parking lots, and gas stations. Moreover, temporary urban planning can enhance the vitality of idle, closed, or uninhabitable urban spaces, enabling them to provide new services and undergo transformation (Fig 3d).

## The evolution of the original sites of fortifications in Beijing

### The demolition of the Ming and Qing Dynasties' city walls and the planning of the second ring road

The City Walls of the Ming and Qing Dynasties in Beijing were reconstructed and expanded from the original walls of the Yuan Dynasty. Construction began in the first year of the Ming Dynasty (Hongwu, 1368) and concluded in the forty-third year of the Ming Dynasty (Jiajing, 1564). They comprised inner and outer walls, a moat, gate towers, and barbicans (Fig 6a). The inner-city wall spanning 23.3 kilometers with nine gates and gate towers, exceeded the outer wall's specifications. The outer wall, measuring 14.41 kilometers with seven gates and gate towers, completed the overall "convex" pattern [28].

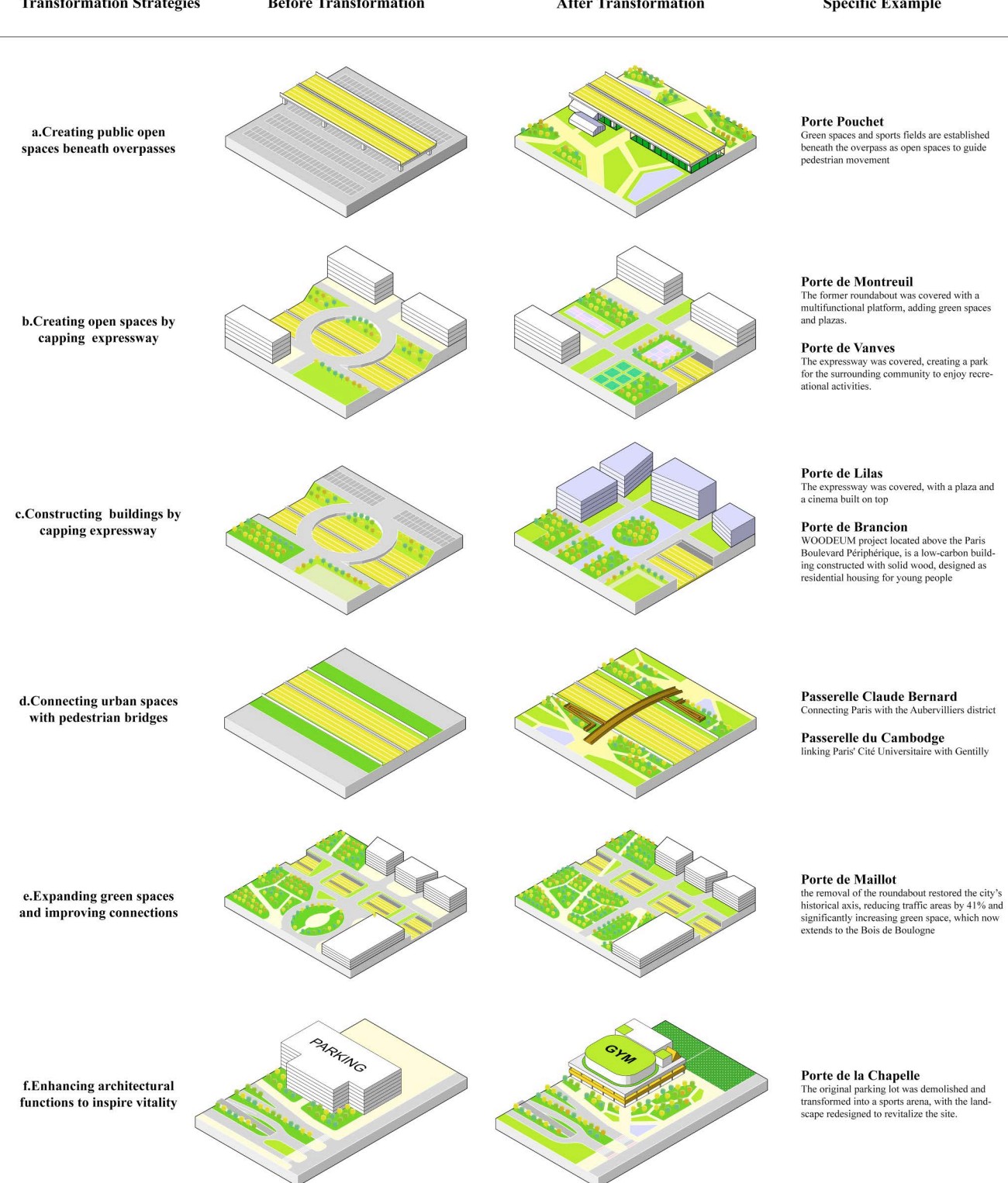

Fig 5. **Strategies for Improvement the Original Sites of Fortifications in Paris.**

Following the establishment of the People's Republic of China in 1949, debates over city wall preservation ensued. Chinese civil engineering expert Hua Nangui advocated demolishing the city walls to eliminate barriers between urban and suburban areas and promote architectural harmony between the interior and exterior of the city, suggesting their foundations be repurposed for road construction [29]. Conversely, renowned Chinese architect Liang Sicheng argued for preserving the Beijing city walls as integral to the ancient city, proposing their transformation into a unique park encircling the city [30]. However, despite ongoing discussions, no definitive conclusion was reached. Additionally, by 1957, owing to neglect and unauthorized removal of bricks, the outer city walls were almost completely dismantled [31].

The "Great Leap Forward" movement politically stigmatized the city walls. In September 1958, the Urban Planning Committee's "Master Plan for Beijing" explicitly stated: "Demolish the city walls and construct the Second Ring Road along the river." Subsequently, the Beijing Municipal Planning Bureau and Architectural Design Institute drafted several detailed planning proposals for the old city, none of which were implemented. The 1958 master plan and the 1962 proposal preserved the moat and transformed the original sites of the fortifications into riverside green spaces, with roadways flanking the moat and green space, completely different from the current layout of the Second Ring Road (Fig 6b). The Beijing municipal government cautiously debated the walls until 1965 when they began dismantling them to build the subway for defense needs. Demolition lasted until 1969.

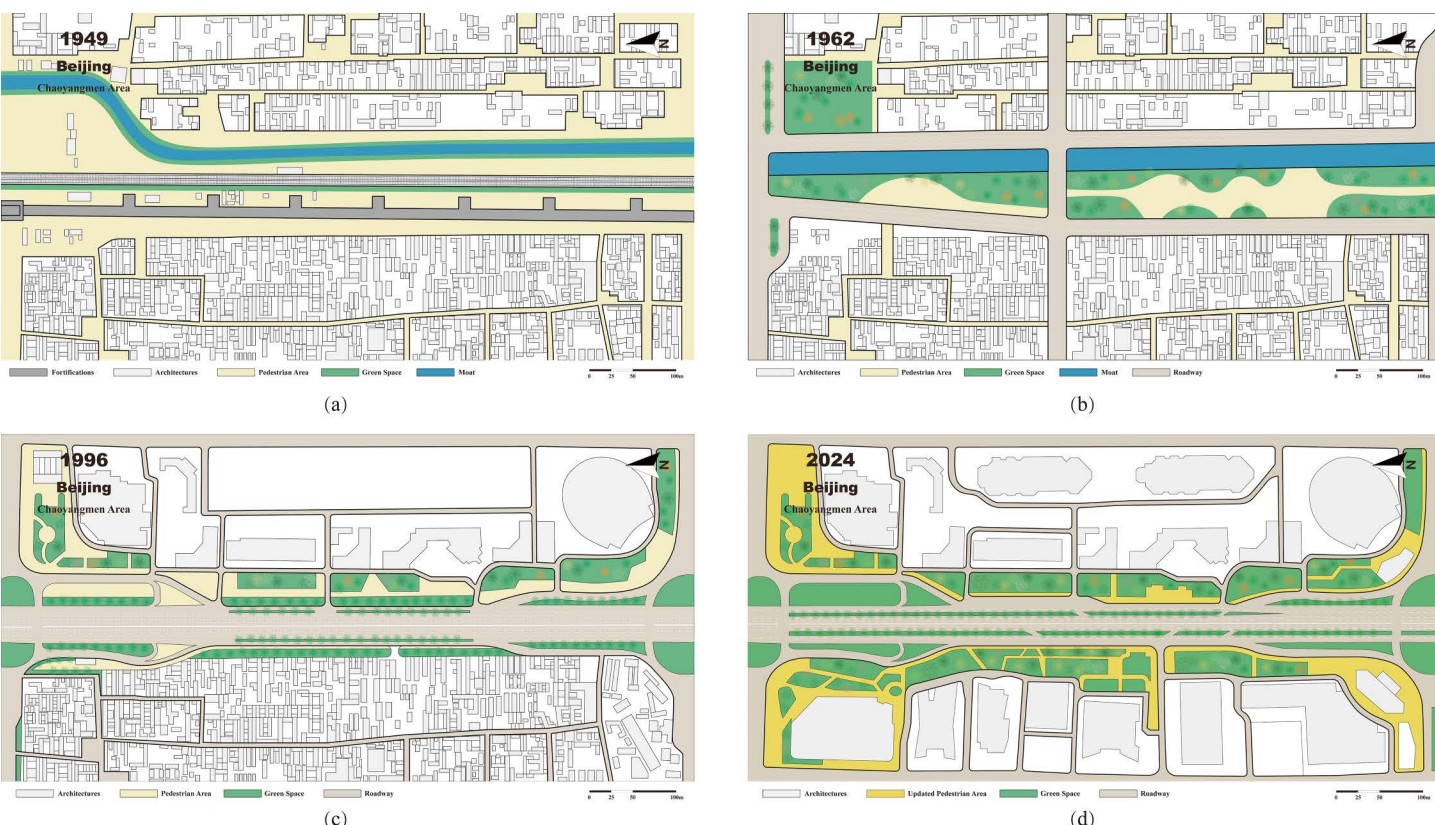

**Fig 6. Evolution of the Original Sites of Fortifications in the Chaoyangmen Area, Beijing:** (a) Plan of the Chaoyangmen area in 1949, before the city walls were demolished; (b) Planning map of the original site of the city walls in the Chaoyangmen area in 1962; (c) Plan of the Chaoyangmen area in 1996, showing the moat being buried and the construction of high-rise buildings beginning on the outer side of the Second Ring Road; (d) Plan of the Chaoyangmen area in 2024, showing a significant increase in green space, with Siheyuan inside the Second Ring Road replaced by high-rise buildings.

## The construction of the second ring road and surrounding Urban Areas

Beijing's city walls were demolished simultaneously with the construction of the subway and the Second Ring Road. Between 1965 and 1971, the inner city walls of Beijing were completely demolished, and by 1969, the first subway line was completed [32]. In 1971, as the subway protection layer was to be used as a planned road surface, "Line 2 of the Beijing Subway Project" and the "Northern Second Ring Road" were implemented simultaneously, with the East and West City Moats converted into underground channels. By the end of 1980, the Northern Second Ring Road, which included nine overpasses, was completed. Line 2 was completed in 1984. In 1987, the "Southern Second Ring Project" was initiated, placing the road outside the moat. In 1992, the entire Second Ring Road was opened to traffic, becoming China's first expressway at the city entrance [33].

In August 1985, Beijing introduced the "Beijing Urban Area Building Height Control Scheme," capping buildings heights in the old city at 45 meters, with certain areas not allowed to exceed 60 meters [34]. However, the completion of the Second Ring Road and the transformation of the economic system fueled the continuous construction of high-rise buildings around it, continually defying urban planning constraints, especially in the East and West Second Ring areas [35]. Some buildings along the East Second Ring Road, such as Galaxy SOHO and the Petroleum Building, are enormous, not only creating a significant sense of oppression for pedestrians but also severely clashing with the surrounding urban image. On the east side of the West Second Ring, Financial Street's buildings are generally approximately 100 meters tall, with the iconic B7 Building reaching 116 meters. This area lacks variation in the skyline and exerts significant pressure on urban traffic owing to its high floor area ratio (Fig 6c).

Overall, the Second Ring Road has catalyzed the development of surrounding areas; however, it has also introduced numerous problems. Thes include overly wide roads, a fully enclosed road system, and complex interchanges that greatly inconvenience pedestrians and cyclists. Building heights surpass planned controls, disrupting the spatial pattern of the old city, transforming it from the original spacious and open layout to the current situation where the exterior is high, and the interior is low. The concentration of high-rise buildings has exacerbated urban traffic congestion, leading to increased air and noise pollution.

## The transformation strategies of the original sites of fortifications in Beijing

By the end of the 20th century, the Beijing municipal government gradually recognized the problems caused by the Second Ring Road, such as environmental pollution, navigation difficulties, and the disruption of the old city layout. Starting in 2000, a theme of "Green City Wall" shape of the Beijing Ming, resulting in the creation of a green belt at least 30 meters wide and several historical and cultural parks. In 2009, Beijing launched the "Second Ring Urban Greenway" project based on the "Green City Wall" concept, integrating the ideas of "urban greenways" and "slow-traffic cities," aiming to create a multifunctional greenway. The "Beijing Master Plan (2016–2035)" suggested constructing a "cultural landscape line along the Second Ring Road" to enhance Beijing's historical and cultural heritage alongside its modern capital façade [36]. Since the 21st century, the planning concept for the original sites of city walls has evolved from basic greening to a focus on ecological and cultural landscapes, with strategies including the sensible utilization of historical and cultural heritage, enhancement of public space, and the establishment and optimization of a slow-traffic system.

First, to protect and restore the remnants of city walls and defense facilities, they should be integrated into urban public spaces to enhance the city's historical and cultural landscape (Fig 7a). Currently, the Deshengmen Embrasured Watchtower, the Zhengyangmen Gate

Tower and Embrasured Watchtower, and their surrounding areas have been transformed into urban squares, and the Dongbianmen Ming Dynasty city wall site has been developed into a park. Additionally, since the beginning of the 21st century, Beijing has begun to restore certain city wall sites. The Yongdingmen Gate Tower was reconstructed as an urban square, and the Zuo'anmen Corner Tower was restored and is now used as a library. Improvement and restoration projects for the moats are progressing in an ordered manner. The North Moat Bank has been transformed into an ecological revetment featuring a slow-traffic system and scenic nodes. The historical water system of the Qiansanmen Moat has also been slated for restoration.

Secondly, to improve the public space environment (Fig 7b). since 2000, guided by the "Green City Wall" concept, Beijing has developed many urban public spaces such as Shuncheng Park, Northern Second Ring Urban Park, and Desheng Park. These are primarily linear urban parks that blend with the surrounding ancient cityscape, accentuating Beijing's history. The open spaces in the Eastern Second Ring Commercial District showcase a modern style integrated with adjacent high-rise buildings, displaying a fashionable, international, and contemporary urban image.

Finally, a slow-traffic system was implemented along both sides of the Second Ring Road, integrating the surrounding historical, cultural, and natural landscapes to construct an urban greenway that merges ecological, health, heritage, and aesthetic elements (Fig 7c). The "Second Ring Urban Greenway" has 25 scenic nodes and links several large parks, such as the

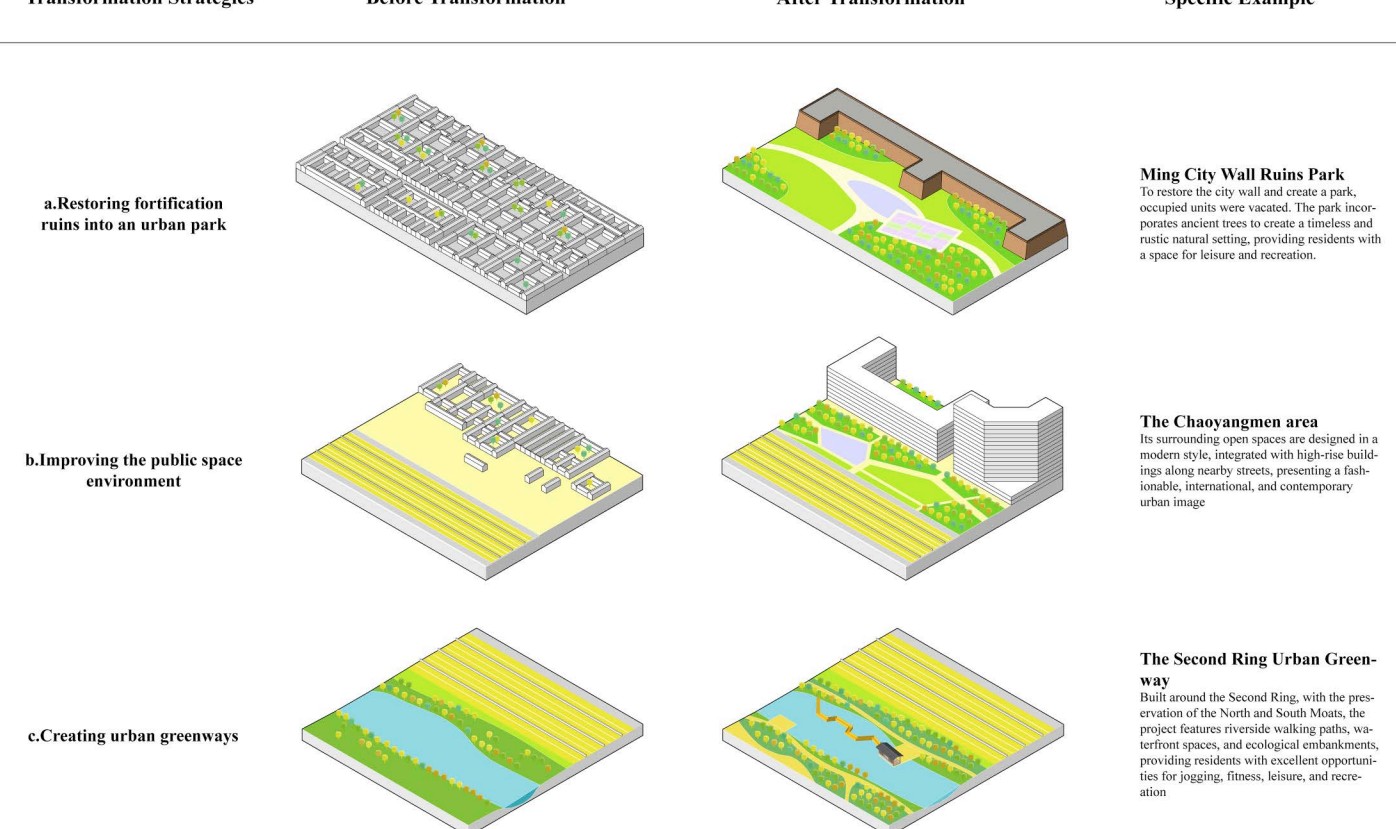

| Transformation Strategies | Before Transformation | After Transformation | Specific Example |
|---|---|---|---|
| **a.Restoring fortification ruins into an urban park** | | | **Ming City Wall Ruins Park** To restore the city wall and create a park, occupied units were vacated. The park incorporates ancient trees to create a timeless and rustic natural setting, providing residents with a space for leisure and recreation. |
| **b.Improving the public space environment** | | | **The Chaoyangmen area** Its surrounding open spaces are designed in a modern style, integrated with high-rise buildings along nearby streets, presenting a fashionable, international, and contemporary urban image |
| **c.Creating urban greenways** | | | **The Second Ring Urban Greenway** Built around the Second Ring, with the preservation of the North and South Moats, the project features riverside walking paths, waterfront spaces, and ecological embankments, providing residents with excellent opportunities for jogging, fitness, leisure, and recreation |

**Fig 7. Strategies for Improvement the Original Sites of Fortifications in Beijing.**

Ming Dynasty City Wall Ruins Park and Tiantan Park. However, greenways face issues such as monotonous walking spaces, insufficient plant arrangements, and underutilized spaces under overpasses. Therefore, the latest proposal "cultural landscape line along the Second Ring Road," aims to enhance these aspects, including connecting slow-traffic systems, improving the greenspace system, and developing a city wall ruin park loop (Fig 6d).

## Comparative analysis of the original sites of fortifications in three cities

To explicitly compare the similarities and differences in the evolution of the original sites of fortifications in Beijing, Paris, and Moscow, we developed a quantitative analytical method based on ArcGIS Pro platform. Spatial distribution maps of the original sites of fortifications during key historical periods were created for the three cities, and the proportional distribution of functional spaces within these sites was calculated across different timeframes (Fig 8). The results reveal that the evolution of the original sites of fortifications in all three cities follows a similar staged pattern, which can be divided into three major phases: the demolition and planning phase, the development and construction phase, and the reflection and renewal phase (Figs 9, 10). Although the time span of these phases varies among the three cities, the processes within each phase show remarkable commonalities. These similarities are primarily attributed to the shared political, economic, and social challenges encountered by the three cities during their respective urbanization processes.

## Phase 1: Demolition of fortifications and planning of their original sites

The first phase involves the demolition of fortifications and the planning of their original sites. During this phase, the three cities exhibited highly similar planning strategies. Specifically, they all removed their fortifications and adopted a combined approach of ring roads and green belts on the original sites of fortifications. Particular attention was paid to the design of green belts, resulting in a significant increase in the proportion of open spaces and a noticeable rise in transportation spaces.

The quantitative analysis shows that in Moscow's 1816 city plan, the central 25 meters of the Zemlyanoy Rampart's original site was designated as a roadway, flanked by public gardens on both sides. This increased the proportion of open spaces to 38.65%, while transportation spaces rose from 26.55% to 46.02%. In Paris, after the fortifications were dismantled in 1919, the 1924 city plan divided the original sites of fortifications into two urban belts comprising social housing and green belts. This increased the proportion of open spaces from 18.83% to 45.39%, with transportation spaces slightly rising to 15.36%. In Beijing, the demolition of fortifications began in 1953, and the 1962 city plan transformed the original sites of fortifications into riverside green belts. Vehicle lanes were placed along both sides of the moat and the green belts, increasing the proportion of open spaces from 29.4% to 63.5%, while transportation spaces rose to 31.73%.

Although the timing of the demolition of fortifications differed among the three cities, all occurred in the aftermath of significant wars. In Moscow, following the Napoleonic Wars in 1812, the destroyed fortifications were not rebuilt. The traditional bastioned walls were deemed ineffective against the long-range artillery of the time, and the growing city required more space for expansion. In contrast, Paris went against this trend in 1844 by constructing the bastioned Thiers Wall. However, this fortification later proved ineffective during the Franco-Prussian War and World War I, leading to its eventual demolition after the wars. In Beijing, the dismantling of the city walls began in 1953, after World War II and the Chinese

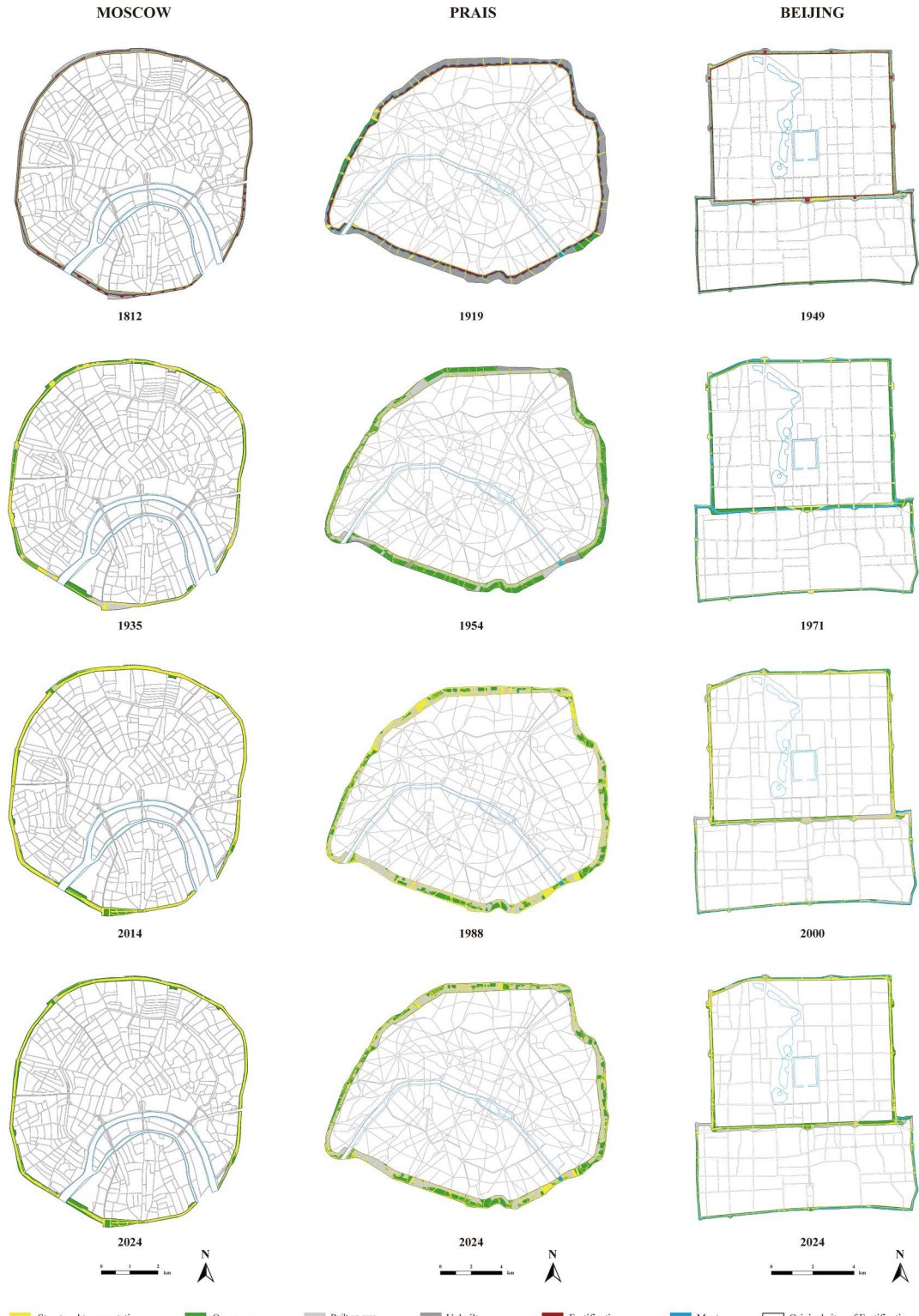

**Fig 8. Functional Distribution Diagram of the Original Sites of Fortifications in Three Cities Across Different Stages.**

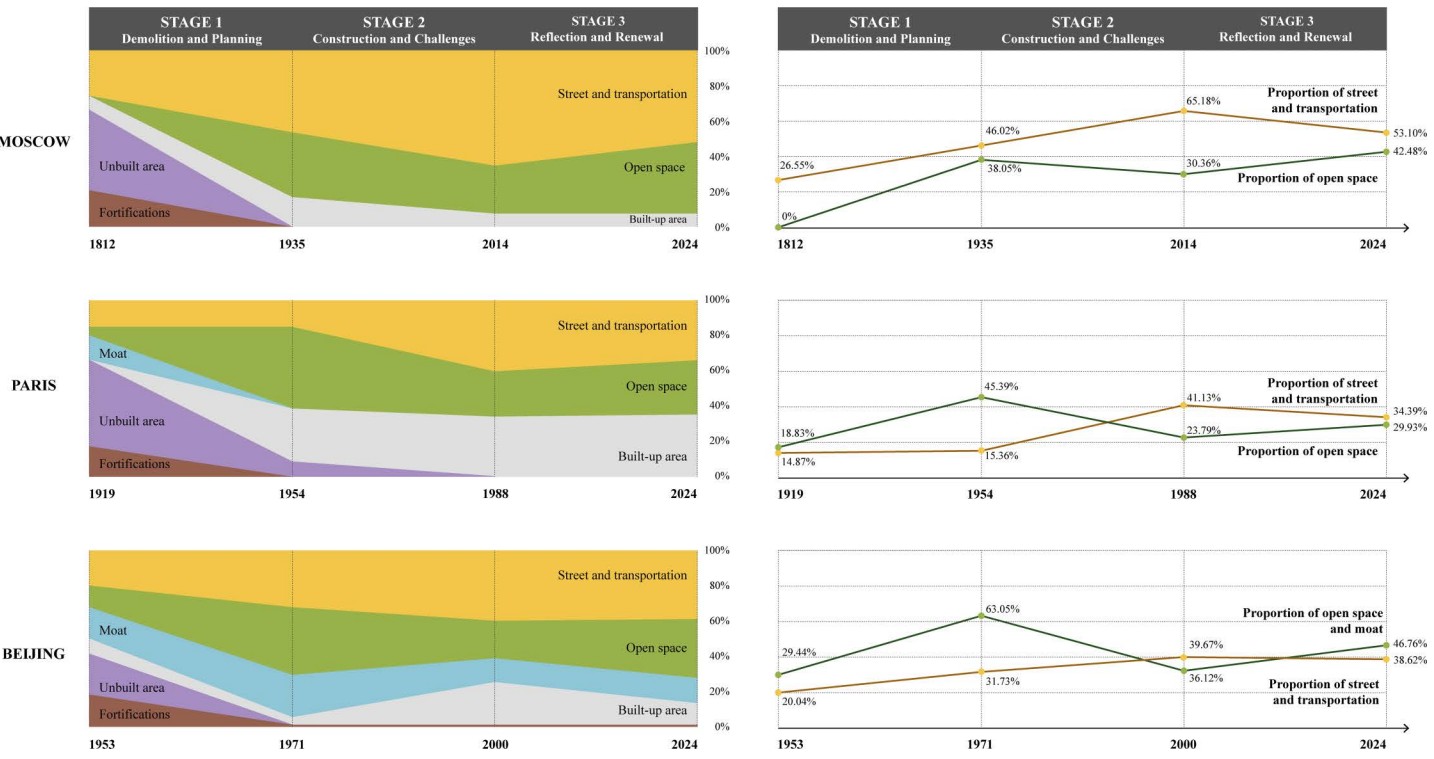

**Fig 9. Diagram of Changes in Internal Functional Proportions of the Original Sites of Fortifications in Three Cities.**

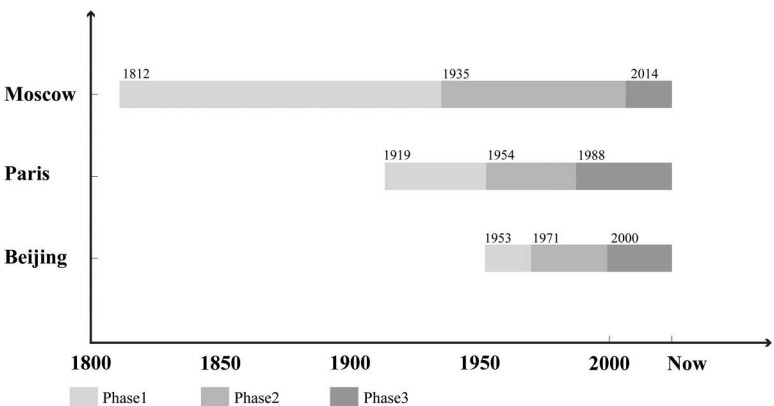

**Fig 10. Time Division of the Three Stages of the Original Sites of Fortifications in Three Cities.**

Civil War. By 1958, the walls were completely removed, partly due to their political association as symbols of feudalism, and to accommodate urban development and improve transportation infrastructure.

Overall, the decision to dismantle fortifications in all three cities reflects the complex socio-political contexts of the post-war periods. These decisions were driven by the pressing need for urban expansion and infrastructure development, as well as political transitions and the evolving symbolism of city walls. While the demolition of these historically and culturally

significant fortifications may seem regrettable from a modern perspective, it was a necessary choice for the development of capital cities under the circumstances of the time.

In the urban planning following the demolition of fortifications, all three cities saw a significant increase in the proportion of open spaces and adopted a combined approach of ring roads and green belts. This practice was partly influenced by the construction of tree-lined boulevards during the reign of Louis XIV in France. Moscow's planning may have incorporated elements of public participation in its design. Additionally, the later demolition of fortifications in Paris and Beijing appears to have been shaped by 19th-century public health and garden city ideologies [37]. These concepts emphasized the importance of increasing green spaces and public areas to enhance residents' health, reduce the spread of diseases, and prevent excessive urban sprawl. Consequently, the urban planning of Paris and Beijing incorporated wider and more spacious open areas in the original sites of their fortifications. These designs aimed to provide high-quality living environments while supporting the orderly development of the cities (Table 1).

## Phase 2: Development and construction of the original sites of fortifications

The second phase marks the development and construction on the original sites of fortifications. During this phase, the functional transformations of the three cities exhibited similar characteristics: a significant increase in transportation spaces, a noticeable reduction in open spaces, and a gradual departure from the planning patterns established in the first phase.

Quantitative analysis shows that Moscow's "Garden Ring" was widened from the original 25 meters to 30–40 meters, sacrificing gardens and iconic buildings along the route. This led to an increase in the proportion of transportation spaces from 46.02% to 65.18%, while open spaces decreased from 38.05% to 30.36%.

In Paris, the construction of new residential buildings and the creation of the 36- to 60-meter-wide "Boulevard Périphérique" significantly reduced the planned green belt area, with open spaces dropping from 45.39% to 23.79%. Simultaneously, transportation spaces increased from 15.36% to 41.13%, and land allocated for buildings rose from 31.02% to 35.08%.

In Beijing, the Second Ring Road, with a width ranging from 50 to 70 meters, further expanded with the addition of overpasses and other facilities. Transportation spaces exceeded the values initially planned, reaching 39.67%, while the originally planned 63.05% of open space was not realized. Instead, the actual open space achieved during construction was reduced to 36.12%.

During the second phase, all three cities shared similar political and economic contexts: stable political conditions, rapid population growth, accelerated industrialization and urbanization, and an urgent need for infrastructure development.

Table 1. Comparison of Basic Information on Fortifications of Three Cities.

|  | Moscow | Paris | Beijing |
|---|---|---|---|
| Construction Time | 1591 | 1841 | 1564 |
| Demolition Time | 1812 | 1919 | 1953 |
| Length of fortification | 15.6km | 35km | 40km |
| Width of fortification | 60—70m | 400m | 80—120m |
| Area of the Old City | 18.2km2 | 74km2 | 60km2 |
| Planned Road Width | 25m | 40m | 60—100m |

Moscow entered this phase the earliest. Since Stalin launched the "First Five-Year Plan" in 1928, the Soviet Union experienced rapid industrialization accompanied by a dramatic population increase, creating an urgent need to expand transportation systems and housing. In 1935, Moscow initiated its General Urban Plan, which included widening the Garden Ring and constructing high-rise buildings along its route. This plan not only met the demands of urbanization but also embodied Stalin's political ambition to showcase socialist modernization achievements through urban development.

Paris entered this phase in 1953. Although France had experienced relatively rapid industrialization during the late 19th and early 20th centuries, it faced economic difficulties following World War II, compounded by population growth and the need to restore infrastructure and industrial capacity. To promote economic recovery and address housing shortages, the French government passed the "Loi Raffarin" in 1953, allocating 20% of the green belt space on the original fortification sites for residential construction. Simultaneously, to alleviate traffic congestion, the government decided in 1954 to build the Boulevard Périphérique, a ring expressway.

Beijing was the last to enter this phase. After the founding of the People's Republic of China in 1949, urbanization and industrialization gradually unfolded. Although the 1958 urban plan proposed the construction of ring roads, economic development was hindered by political events such as the "Great Leap Forward" and the "Cultural Revolution." It was not until 1971, with economic recovery and population growth, that the government formally initiated the construction of the Second Ring Road, incorporating subway development into the plan after the demolition of the fortifications.

Furthermore, during the mid-20th century, the influence of modernism and functionalism, particularly the "Megastructure Thought", profoundly shaped the development of the original sites of fortifications. This concept advocated for large-scale, integrated architectural and infrastructure systems to address urbanization challenges. Under this paradigm, wide ring roads were conceived as "large-scale infrastructure" projects that not only effectively connected urban regions and improved traffic flow but also supported the rapid spatial expansion of cities. This approach reflected a strong emphasis on functional integration and efficiency, becoming a defining characteristic of urban development during this phase (Table 2).

## Phase 3: Reflection and renewal of the original sites of fortifications

The third phase is characterized by the reflection and renewal of the original sites of fortifications. During this stage, the functional evolution of Beijing, Paris, and Moscow revealed significant commonalities: the proportion of open spaces generally increased, transportation

**Table 2. Comparison of Urban Ring Roads Built on the Original Sites of Fortifications in Three Cities.**

|  | Moscow | Paris | Beijing |
|---|---|---|---|
| Construction Time | 1935—1965 | 1954—1973 | 1971—1992 |
| Road Width | 60—70m | 35—60m | 50—70m |
| Road Type | Significant traffic route | Enclosed expressways | Enclosed expressways |
| Road Design | Mainly flat terrain, with intersections as overpasses | 50% underground or tunnel, 40% elevated, 10% at grade | Mainly flat terrain, with intersections as overpasses |
| Differences from Preliminary Planning | Gardens on the Ring Road have disappeared, along with some iconic buildings | Parts of the green belt have been used for the construction of the Boulevard Périphérique, social housing, and infrastructure | The moat has been covered, and the original green spaces have been converted into the Second Ring Road. |

spaces decreased, and urban planning began to return to the original design concepts of earlier phases.

Quantitative analysis shows that Moscow expanded sidewalks and green spaces through traffic optimization and the reshaping of key landscape nodes, increasing the proportion of open spaces from 30.36% to 42.48%, while transportation spaces decreased from 65.18% to 53.10%. In Paris, the covering of expressways and the redevelopment of green landscapes reduced transportation spaces from 41.13% to 34.39% and increased open spaces from 23.79% to 29.93%. Beijing, by utilizing its historical and cultural heritage, established landscape nodes and slow-traffic greenways, increasing the proportion of open spaces from 36.1% to 46.76%.

During this phase, urbanization in the three cities stabilized, and planners began to reflect on the problems caused by previous car-centric urban development models, such as environmental pollution, spatial fragmentation, and the loss of historical continuity. Influenced by globalization, concepts such as sustainable development, cultural heritage preservation, human-centered urbanism, and ecological balance profoundly shaped each city. These ideas gave rise to concepts like livable cities, green cities, walkable cities, and the 15-minute city, which guided the systematic rethinking of challenges associated with the original sites of fortifications and the formulation of targeted solutions.

Since 1988, Paris was the first to initiate the renewal and redevelopment of its original fortification sites. Through measures such as expressway coverage, enhanced regional connectivity, functional adjustments, and interim urban planning, Paris significantly improved the vitality and efficiency of these areas. Particularly in the areas surrounding the former city gates, Paris implemented detailed and continuous urban design, enhancing spatial quality and functionality. Moscow adopted a simpler yet effective approach by optimizing traffic layouts and widening sidewalks, improving the accessibility and user experience of public spaces. In contrast, Beijing, by preserving parts of its city wall relics and the north-south moats, emphasized the protection and utilization of historical and cultural heritage in its renewal strategies, reflecting a balanced approach to integrating historical preservation with modern urban development.

## Discussion and conclusion

The original sites of fortifications, as part of urban historical landscapes, demonstrate the layered integration of urban functions and forms across different historical periods. According to the Historic Urban Landscape (HUL) theory, urban heritage is characterized by stratification, where historical layers collectively contribute to its overall value. Based on this framework, this study combined qualitative and quantitative approaches to examine the evolution of the original sites of fortifications in Paris, Beijing, and Moscow. The qualitative analysis identified key historical events and contexts that drove significant changes to the sites, while the quantitative analysis, conducted using the ArcGIS Pro platform, measured functional changes in these sites over different time periods. The findings reveal that, despite differences in the timing of major transformations, all three cities experienced three distinct phases of evolution: demolition and planning, development and construction, and reflection and renewal. The underlying driving forces behind these phases exhibited significant similarities.

The first phase occurred during the early stages of urbanization, often following major wars and during a "golden period" of population growth. In this phase, as urbanization accelerated, many first-tier cities globally, driven by political, economic, and social factors, dismantled their fortifications. Influenced by garden city planning principles, they typically adopted designs that combined ring roads with green belts. These designs often commemorated the historical presence of the fortifications by transforming their original sites into open spaces.

The second phase coincided with the most rapid period of urbanization, during which functionalism became the dominant ideology in urban planning. To meet the rapidly growing demand for transportation, ring roads were designed excessively wide, resulting in a substantial increase in transportation spaces and a significant reduction in green spaces. The expansion of ring roads introduced a series of problems, including environmental pollution, ecological degradation, and spatial fragmentation. Increased vehicle traffic and emissions worsened air quality, while the lack of pedestrian and cycling systems made non-motorized travel unsafe. Moreover, ring roads became physical barriers, making it difficult for pedestrians to cross, and further fragmented the urban spaces on either side. Ironically, these ring roads evolved into new "walls" within the cities.

The third phase emerged during a period of slowed urbanization, as large-scale construction projects gave way to a focus on urban stock development. During this phase, there was increasing reflection on the adverse impacts of overdevelopment on the original sites of fortifications, including ecological degradation, environmental deterioration, and urban fragmentation. At the same time, evolving societal demands and values—emphasizing sustainability, human-centered urbanism, ecological balance, and cultural heritage preservation—profoundly influenced urban planning. In response, the three cities implemented measures to address these challenges. During this phase, the proportion of transportation spaces began to decrease, while open spaces started to recover.

The stratified analysis of the original sites of fortifications reveals that, despite differences in the social contexts and historical conditions of the three cities, their approaches to addressing challenges related to these sites exhibit similar logical patterns. These strategies reflect a degree of universality in addressing the challenges faced by cities globally. The key findings can be summarized as follows:

(1) Prioritizing ecological continuity and sustainability: Designing greenways in the original sites of fortifications ensures ecological continuity and sustainability. This approach not only encourages low-carbon transportation and promotes public health but also preserves and perpetuates the historical legacy of the fortifications within modern urban spaces.

(2) Enhancing spatial connectivity and social integration: Urban planners can strengthen the connectivity of urban areas on both sides of the original sites through the design of open spaces, pedestrian bridges, walkways, and buildings. These interventions "stitch" the fragmented urban spaces and foster regional integration and social interaction, preventing ring roads from becoming urban "chasms."

(3) Incorporating diverse functions to activate spaces: Integrating cultural, commercial, recreational, and community service functions into the original sites of fortifications can transform these areas into vibrant urban spaces, preventing them from becoming "lost spaces."

Despite the contributions of this study, several limitations remain. First, the case studies are limited to three cities. While these cities are representative, the findings may not fully apply to cities with different cultural contexts and developmental stages. Second, the quantitative analysis relies heavily on the ArcGIS Pro platform and is constrained by the completeness and accuracy of historical data. As a result, functional changes during certain periods could only be indirectly inferred. Furthermore, the stratified historical analysis lacks depth in linking specific details to historical events, which limits its ability to fully capture the dynamic changes and their impacts across different periods. Future research should expand the scope of study and incorporate multi-dimensional datasets to enhance the dynamic analysis of the original sites of fortifications. These efforts would provide more comprehensive theoretical and practical guidance for heritage preservation and sustainable urban development.

## Supporting information

**S1 File. Land use of the original sites of fortifications.**
(XLSX)

**S1 Data. Functional distribution maps of the original sites of fortifications in Beijing, Paris, and Moscow.**
(ZIP)

## Author contributions

**Conceptualization:** Mo Xu.

**Data curation:** Mo Xu, Haolin Zhu.

**Formal analysis:** Mo Xu.

**Methodology:** Mo Xu, Haolin Zhu.

**Resources:** Mo Xu.

**Software:** Mo Xu.

**Validation:** Mo Xu, Haolin Zhu.

**Visualization:** Mo Xu.

**Writing – original draft:** Mo Xu, Haolin Zhu.

**Writing – review & editing:** Mo Xu, Haolin Zhu.

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
