## [Decision Letter · Decision Letter 0]

29 Oct 2024

PONE-D-24-28746A Study on the Evolution of Original Sites of Fortifications from the Perspective of HUL: Cases of Paris, Beijing, and MoscowPLOS ONE

Dear Dr. Xu,

Thank you for submitting your manuscript to PLOS ONE. After careful consideration, we feel that it has merit but does not fully meet PLOS ONE’s publication criteria as it currently stands. Therefore, we invite you to submit a revised version of the manuscript that addresses the points raised during the review process. The reviewers' comments are as follows;

**Reviewer 1**

This manuscript presents an important contribution to the field of urban studies and offers valuable insights into the historical and contemporary significance of fortification sites. The suggestions are as follows:

The term "stratification process" is an important concept but has not been clearly defined in the manuscript. It is recommend to providing a concise explanation of what this entails in the context of fortifications and urban landscapes.

The current literature review could be expanded to better situate your findings in the broader academic discourse. Discussing previous studies on urban fortifications and their transformations will demonstrate the novelty of your research and highlight existing gaps that your work addresses.

Methodology: How were the data collected? What criteria were used for the selection of case studies? It is recommend to including this information to improve the reliability of the research.

Consider exploring how specific socio-political and historical contexts in each city shaped the unique outcomes of their fortification sites. These differences could provide deeper insights into the stratification processes you describe. In the discussion part, the authors could discuss more about how can the lessons from these three cities be applied to similar projects in other cities?

**Reviewer 2**

The manuscript provides a comprehensive perspective to understand the similarities and differences of different cities in dealing with historical fortification sites by comparing three cities under different cultural and political backgrounds, and provides meaningful experience for the protection and development of fortification sites in other cities. However, the manuscript still needs to be improved in the following aspects:

1.The reason for dividing into three stages is not clearly explained. Why are these three periods? Judging from Figure 14, the time spans of these three fortifications are very different. Moscow's fortifications span 200 years, while Beijing's only has 50 years. Their phases are the same. What is the reason behind this?

2.The manuscript, while providing a comprehensive perspective, lacks a crucial element-a three-dimensional analysis of the fortifications of the three cities. The urban landscape, with its planar and three-dimensional changes, is a key aspect of understanding urban history. It is recommended to add an analysis of the three-dimensional aspect through photos or 3D models as this will significantly enhance the readers' understanding of the subject.

3.The discussion of strategies in each period is relatively general. To truly understand the differences in strategies among the three cities, it is essential to have specific strategy analysis diagrams. These could include analyzing the changes in city walls, roads, greenery, and pedestrian spaces from a sectional perspective. Figure 15, while informative, is also quite general and does not clearly depict the differences in strategies. Therefore, the addition of specific strategy analysis diagrams is recommended to enhance the readers' understanding.

4.Minor error: On line 491, "Comparative Analysis of the Three Cities in Phase Two", it should be Phase Three.

**Reviewer 3**

Many typographical errors throughout the entire document. Odd spacing, parts of words, and inconsistent capitalizations.

Although the research is solid, the main premise of the research, beautiful city, is not clearly laid out. No background, examples, or definitions are provided, only general terms about harmonious development. It is unclear if this is a strategic design agenda (a Beautiful City initiative) or just a category developed by the authors (generally designing beautiful cities). This needs to be corrected for the findings and conclusions to be accurately assessed. Currently, there is no clear description of the “beautiful city” development practices.

Additionally, tourism is described as “green, low-carbon, and environmentally friendly nature of tourism” in the last paragraph of page six. This is a broad assumption with no supporting data. Furthermore, one of the ways tourism and urban development is later assessed is the increase in car ownership. This creates a paradox - - how can tourism be “green” but it’s success be measured on the increase in car-ownership?

Lastly, the overall purpose of the research is unclear. What is the intention of this coupling model? There is no clear need or application offered.

**Reviewer 4**

Comments and Suggestions for Authors

Based on the perspective of HUL, this paper discusses the evolution of original sites of fortifications in three cities. The paper is well written and has novel ideas, but there are some areas where the clarity and accuracy of scientific reporting could be improved:

Abstract and Introduction: An explanation of the results can be appropriately included in the summary, so that readers can get the relevant information more quickly. The introduction may appropriately set the background of the fortifications, but additional explanations of key theories and frameworks can enhance the reader's understanding, especially for interdisciplinary readers unfamiliar with the subject.

Text description: The overall structure of the article can be more clear. For example, in the case analysis of each city, a unified analytical framework can be adopted, such as following the sequence of "historical evolution of the fortress - demolition and planning - construction process and problems - reflection and renewal strategies", so that readers can compare the similarities and differences between different cities more easily.

Methodology: This paper studies the historical landscape of the city from HUL's novel perspective, but it does not clearly point out how the method is realized. At the same time, the frequency of HUL mentioned in the whole paper is low.

Results Presentation: The overall structure of the article is rigorous, but suggestions for city builders can be added in the conclusion part by summarizing these three case cities.

Discussion and Conclusion: When discussing the similarities and differences of the evolution of the three urban fortress sites, the depth of analysis can be further deepened. For example, in the common part, in addition to pointing out similar stages and planning methods, we can also explore the deep-rooted reasons behind these commonalities, such as the macro trend of global urbanization process, and the general influence mechanism of similar urban development needs (such as traffic improvement, population accommodation, etc.) on the evolution of fortress sites.

Scientific Language: Overall, the language is formal and suitable for a scientific audience. But some parts of the paper, particularly in the literature review and conclusion, could be more concise.

Limitations: The study identified some key limitations, such as not having enough long-term, continuous data, which is important for scientific transparency. Expanding the generality of how these limitations affect the findings will strengthen the paper.

**Reviewer 5**

Dear authors,

Thank you very much for the opportunity to read your paper. From a general perspective your topic is a real interest one, even the research is mostly at a descriptive level. So, to increase the consistency of your paper I recommend to include some statistics about the profile of these alpha cities (population, PIB, density etc), preferably with a global perspective (the rank of each city by various criteria into a global score).

Also, please review carefully your text, I suppose that at 491 it is actually Phase Three and not again Two (as it is at 318).

**Reviewer 6**

The article examines the transformation of historic fortifications in Paris, Beijing, and Moscow over time. While initial plans included green and walkable spaces, actual deconstruction (of fortified area) and their redevelopment often sacrificed those plans to provide for automobiles. This led to typical urban ills including increasing pollution, decreasing connectivity and so forth. However, all three of these cities are now in the process of rethinking their approaches to these historic sites.

Overall, this article is an engaging read and presents an interesting comparison of three phased processes in three major cities of the world. The analysis of the parallels between the cities is fascinating but I was left wondering how they diverged. I think more engagement with that as well as the broader planning history literature is needed. Also, the crux of the analysis focuses on one smallish area per city, yet no mention is given to how that particular area within each city was chosen (apologies if I missed it) nor is evidence provided as to how representative each particular area is to the changes occurring along the rest of the fortification areas in the cities.

Major comments:

• Lines 1-57 seem to be lacking references. Same for lines 87-94. Same for lines 190-204. More engagement with the existing literature is needed throughout.

• I think it would be useful to further the discussion of the economic, military and political reasons for dismantling the fortifications. This could perhaps be in the form of a table with the three sites as columns, and econ, military, and political reasons as rows with bullet points for each row/sites.

• In Figures 9 and 12, I think it would be helpful to label the streets (especially Blvd des Marechaux and the Périphérique). Similarly, any street or building mentioned in the text, should ideally be labeled on the respective figures to aid in orienting the reader who might not be familiar with the layout of all three of these cities especially on such a fine scale.

• I think the paper would benefit from having a series of pictures of the modern day setting of the three areas of focus (from Google Street View perhaps). It would help situate statements such as this one “Galaxy SOHO and the Petroleum Building, are enormous, not only creating a significant sense of oppression for pedestrians but also severely clashing with the surrounding urban image”

• It is difficult to compare the figures to each other and see the evolution of the different land uses over time. Is there a way to combine these figures so that they are more easily comparable? Also, since these look like GIS files, could you calculate the area devoted to each land use for each city and graph it over time (as a new figure)? Ideally, this would be both for the three smallish areas you focus on and the entire fortified ring area.

• To synthesize your research, maybe you should talk about the process of place making around the fortifications that has taken a long time but that seems to be coming to fruition (finally – after a few setbacks), but how under the guide of the HUL the fortified areas have evolved into pedestrian/bike networks but also important from recreation, environmental sustainability issue as well as cultural stand points (although I’m not sure that was the case in all three cities- perhaps something to discuss in more detail).

• Bringing the discussion back to the HUL is needed in the conclusions.

• I don’t think you should have an abbreviation in the title.

Minor comments:

• Lines 95-104 instead of “this study conduct”, “It analyzes” and other instances, consider saying “we conducted”, “we analyzed”.

• Fig 1. Consider increasing the font on the legend (and a little on the scale bar). Also, I think it would be great to have an inset map that shows a close up (and extent rectangle) of where the sites you focus on in subsequent figures are located relative to the cities themselves, i.e. where is Porte de Vanve in Paris, etc.

• L104, the mention of Figure 1 here seems misplaced. Maybe add it early on when you justify the focus on those three cities?

• Throughout, the periods should come after the parenthetical references, for example, “(10).”

• Figures 4 and 5, since green spaces are the light green, and you are showing trees as the small darker green stars (I’m assuming although that needs to be added to the legend), it’s a bit confusing to have light green with dark green dots as No-building zones (unless you mean parks, or vegetated walkable spaces but then please clarify)?

• When you first mention Hua Nangui and Liang Sicheng, please state who they are (pardon my ignorance).

• Table 1, “construction time” is the start of the construction? Because it took more than a year (at least for Paris), no?

• L223 (but check for other instances for ex L231) the “the” is missing before “Garden ring”

• Also, L226, the verb is missing for “new bridges over the Moscow River” (probably “were built”)

• Lastly, I think that the Moscow River is actually (even in English) referred to as the Moskva River. Similarly, Boulevards of Marshals is called the Boulevard des Maréchaux. I would be inclined to suggest it might be best not to translate all these place names.

• L227, do you mean “in and around the garden rings”?

• L235 “Owing to the Garden Ring’s excessive number of lanes, some sections have up to eight lanes, encroaching on the city’s public space” this sentence needs to be rephrased.

• Somewhere around L260 I think it would be beneficial to mention that the Périphérique is just on the outside of the former fortifications (nearly adjacent to it and essentially marks the boundary between the City of Paris and the suburbs) and that the Périphérique is the reason for the difficulties in going from the City of Paris to the suburbs; not the former fortifications and roads that are within that zone.

• L312, typo “These”

• Consider combining Figures 14 and 15 into one figure as you have a lot of figures and I think it would help comprehension and legibility.

• L384-385 “a 14-kilometer noise barrier” around what?

• L386, what area exactly is “the Crown”?

• L414, add “as well as” between “continuity. Increase”

• L420, maybe specify that the V stands for Vélo (or Bicycle) and not the roman numeral five.

We look forward to receiving your revised manuscript.

Kind regards,

Samuel Kofi Tchum, Ph.D.

Academic Editor

PLOS ONE

Journal Requirements:

2. We note that Figures 1-13 in your submission contain [map/satellite] images which may be copyrighted. All PLOS content is published under the Creative Commons Attribution License (CC BY 4.0), which means that the manuscript, images, and Supporting Information files will be freely available online, and any third party is permitted to access, download, copy, distribute, and use these materials in any way, even commercially, with proper attribution. For these reasons, we cannot publish previously copyrighted maps or satellite images created using proprietary data, such as Google software (Google Maps, Street View, and Earth). For more information, see our copyright guidelines: http://journals.plos.org/plosone/s/licenses-and-copyright.

a. You may seek permission from the original copyright holder of Figures 1-13 to publish the content specifically under the CC BY 4.0 license.  

Additional Editor Comments:

Reviewer 1

This manuscript presents an important contribution to the field of urban studies and offers valuable insights into the historical and contemporary significance of fortification sites. The suggestions are as follows:

The term "stratification process" is an important concept but has not been clearly defined in the manuscript. It is recommend to providing a concise explanation of what this entails in the context of fortifications and urban landscapes.

The current literature review could be expanded to better situate your findings in the broader academic discourse. Discussing previous studies on urban fortifications and their transformations will demonstrate the novelty of your research and highlight existing gaps that your work addresses.

Methodology: How were the data collected? What criteria were used for the selection of case studies? It is recommend to including this information to improve the reliability of the research.

Consider exploring how specific socio-political and historical contexts in each city shaped the unique outcomes of their fortification sites. These differences could provide deeper insights into the stratification processes you describe. In the discussion part, the authors could discuss more about how can the lessons from these three cities be applied to similar projects in other cities?

Reviewer 2

The manuscript provides a comprehensive perspective to understand the similarities and differences of different cities in dealing with historical fortification sites by comparing three cities under different cultural and political backgrounds, and provides meaningful experience for the protection and development of fortification sites in other cities. However, the manuscript still needs to be improved in the following aspects:

1.The reason for dividing into three stages is not clearly explained. Why are these three periods? Judging from Figure 14, the time spans of these three fortifications are very different. Moscow's fortifications span 200 years, while Beijing's only has 50 years. Their phases are the same. What is the reason behind this?

2.The manuscript, while providing a comprehensive perspective, lacks a crucial element-a three-dimensional analysis of the fortifications of the three cities. The urban landscape, with its planar and three-dimensional changes, is a key aspect of understanding urban history. It is recommended to add an analysis of the three-dimensional aspect through photos or 3D models as this will significantly enhance the readers' understanding of the subject.

3.The discussion of strategies in each period is relatively general. To truly understand the differences in strategies among the three cities, it is essential to have specific strategy analysis diagrams. These could include analyzing the changes in city walls, roads, greenery, and pedestrian spaces from a sectional perspective. Figure 15, while informative, is also quite general and does not clearly depict the differences in strategies. Therefore, the addition of specific strategy analysis diagrams is recommended to enhance the readers' understanding.

4.Minor error: On line 491, "Comparative Analysis of the Three Cities in Phase Two", it should be Phase Three.

Reviewer 3

Many typographical errors throughout the entire document. Odd spacing, parts of words, and inconsistent capitalizations.

Although the research is solid, the main premise of the research, beautiful city, is not clearly laid out. No background, examples, or definitions are provided, only general terms about harmonious development. It is unclear if this is a strategic design agenda (a Beautiful City initiative) or just a category developed by the authors (generally designing beautiful cities). This needs to be corrected for the findings and conclusions to be accurately assessed. Currently, there is no clear description of the “beautiful city” development practices.

Additionally, tourism is described as “green, low-carbon, and environmentally friendly nature of tourism” in the last paragraph of page six. This is a broad assumption with no supporting data. Furthermore, one of the ways tourism and urban development is later assessed is the increase in car ownership. This creates a paradox - - how can tourism be “green” but it’s success be measured on the increase in car-ownership?

Lastly, the overall purpose of the research is unclear. What is the intention of this coupling model? There is no clear need or application offered.

Reviewer 4

Comments and Suggestions for Authors

Based on the perspective of HUL, this paper discusses the evolution of original sites of fortifications in three cities. The paper is well written and has novel ideas, but there are some areas where the clarity and accuracy of scientific reporting could be improved:

Abstract and Introduction: An explanation of the results can be appropriately included in the summary, so that readers can get the relevant information more quickly. The introduction may appropriately set the background of the fortifications, but additional explanations of key theories and frameworks can enhance the reader's understanding, especially for interdisciplinary readers unfamiliar with the subject.

Text description: The overall structure of the article can be more clear. For example, in the case analysis of each city, a unified analytical framework can be adopted, such as following the sequence of "historical evolution of the fortress - demolition and planning - construction process and problems - reflection and renewal strategies", so that readers can compare the similarities and differences between different cities more easily.

Methodology: This paper studies the historical landscape of the city from HUL's novel perspective, but it does not clearly point out how the method is realized. At the same time, the frequency of HUL mentioned in the whole paper is low.

Results Presentation: The overall structure of the article is rigorous, but suggestions for city builders can be added in the conclusion part by summarizing these three case cities.

Discussion and Conclusion: When discussing the similarities and differences of the evolution of the three urban fortress sites, the depth of analysis can be further deepened. For example, in the common part, in addition to pointing out similar stages and planning methods, we can also explore the deep-rooted reasons behind these commonalities, such as the macro trend of global urbanization process, and the general influence mechanism of similar urban development needs (such as traffic improvement, population accommodation, etc.) on the evolution of fortress sites.

Scientific Language: Overall, the language is formal and suitable for a scientific audience. But some parts of the paper, particularly in the literature review and conclusion, could be more concise.

Limitations: The study identified some key limitations, such as not having enough long-term, continuous data, which is important for scientific transparency. Expanding the generality of how these limitations affect the findings will strengthen the paper.

Reviewer 5

Dear authors,

Thank you very much for the opportunity to read your paper. From a general perspective your topic is a real interest one, even the research is mostly at a descriptive level. So, to increase the consistency of your paper I recommend to include some statistics about the profile of these alpha cities (population, PIB, density etc), preferably with a global perspective (the rank of each city by various criteria into a global score).

Also, please review carefully your text, I suppose that at 491 it is actually Phase Three and not again Two (as it is at 318).

Reviewer 6

The article examines the transformation of historic fortifications in Paris, Beijing, and Moscow over time. While initial plans included green and walkable spaces, actual deconstruction (of fortified area) and their redevelopment often sacrificed those plans to provide for automobiles. This led to typical urban ills including increasing pollution, decreasing connectivity and so forth. However, all three of these cities are now in the process of rethinking their approaches to these historic sites.

Overall, this article is an engaging read and presents an interesting comparison of three phased processes in three major cities of the world. The analysis of the parallels between the cities is fascinating but I was left wondering how they diverged. I think more engagement with that as well as the broader planning history literature is needed. Also, the crux of the analysis focuses on one smallish area per city, yet no mention is given to how that particular area within each city was chosen (apologies if I missed it) nor is evidence provided as to how representative each particular area is to the changes occurring along the rest of the fortification areas in the cities.

Major comments:

• Lines 1-57 seem to be lacking references. Same for lines 87-94. Same for lines 190-204. More engagement with the existing literature is needed throughout.

• I think it would be useful to further the discussion of the economic, military and political reasons for dismantling the fortifications. This could perhaps be in the form of a table with the three sites as columns, and econ, military, and political reasons as rows with bullet points for each row/sites.

• In Figures 9 and 12, I think it would be helpful to label the streets (especially Blvd des Marechaux and the Périphérique). Similarly, any street or building mentioned in the text, should ideally be labeled on the respective figures to aid in orienting the reader who might not be familiar with the layout of all three of these cities especially on such a fine scale.

• I think the paper would benefit from having a series of pictures of the modern day setting of the three areas of focus (from Google Street View perhaps). It would help situate statements such as this one “Galaxy SOHO and the Petroleum Building, are enormous, not only creating a significant sense of oppression for pedestrians but also severely clashing with the surrounding urban image”

• It is difficult to compare the figures to each other and see the evolution of the different land uses over time. Is there a way to combine these figures so that they are more easily comparable? Also, since these look like GIS files, could you calculate the area devoted to each land use for each city and graph it over time (as a new figure)? Ideally, this would be both for the three smallish areas you focus on and the entire fortified ring area.

• To synthesize your research, maybe you should talk about the process of place making around the fortifications that has taken a long time but that seems to be coming to fruition (finally – after a few setbacks), but how under the guide of the HUL the fortified areas have evolved into pedestrian/bike networks but also important from recreation, environmental sustainability issue as well as cultural stand points (although I’m not sure that was the case in all three cities- perhaps something to discuss in more detail).

• Bringing the discussion back to the HUL is needed in the conclusions.

• I don’t think you should have an abbreviation in the title.

Minor comments:

• Lines 95-104 instead of “this study conduct”, “It analyzes” and other instances, consider saying “we conducted”, “we analyzed”.

• Fig 1. Consider increasing the font on the legend (and a little on the scale bar). Also, I think it would be great to have an inset map that shows a close up (and extent rectangle) of where the sites you focus on in subsequent figures are located relative to the cities themselves, i.e. where is Porte de Vanve in Paris, etc.

• L104, the mention of Figure 1 here seems misplaced. Maybe add it early on when you justify the focus on those three cities?

• Throughout, the periods should come after the parenthetical references, for example, “(10).”

• Figures 4 and 5, since green spaces are the light green, and you are showing trees as the small darker green stars (I’m assuming although that needs to be added to the legend), it’s a bit confusing to have light green with dark green dots as No-building zones (unless you mean parks, or vegetated walkable spaces but then please clarify)?

• When you first mention Hua Nangui and Liang Sicheng, please state who they are (pardon my ignorance).

• Table 1, “construction time” is the start of the construction? Because it took more than a year (at least for Paris), no?

• L223 (but check for other instances for ex L231) the “the” is missing before “Garden ring”

• Also, L226, the verb is missing for “new bridges over the Moscow River” (probably “were built”)

• Lastly, I think that the Moscow River is actually (even in English) referred to as the Moskva River. Similarly, Boulevards of Marshals is called the Boulevard des Maréchaux. I would be inclined to suggest it might be best not to translate all these place names.

• L227, do you mean “in and around the garden rings”?

• L235 “Owing to the Garden Ring’s excessive number of lanes, some sections have up to eight lanes, encroaching on the city’s public space” this sentence needs to be rephrased.

• Somewhere around L260 I think it would be beneficial to mention that the Périphérique is just on the outside of the former fortifications (nearly adjacent to it and essentially marks the boundary between the City of Paris and the suburbs) and that the Périphérique is the reason for the difficulties in going from the City of Paris to the suburbs; not the former fortifications and roads that are within that zone.

• L312, typo “These”

• Consider combining Figures 14 and 15 into one figure as you have a lot of figures and I think it would help comprehension and legibility.

• L384-385 “a 14-kilometer noise barrier” around what?

• L386, what area exactly is “the Crown”?

• L414, add “as well as” between “continuity. Increase”

• L420, maybe specify that the V stands for Vélo (or Bicycle) and not the roman numeral five.

Reviewers' comments:

Reviewer's Responses to Questions

**Comments to the Author**

1. Is the manuscript technically sound, and do the data support the conclusions?

Reviewer #1: Yes

Reviewer #2: Yes

Reviewer #3: Yes

Reviewer #4: Yes

Reviewer #5: Yes

Reviewer #6: Partly

2. Has the statistical analysis been performed appropriately and rigorously? 

Reviewer #1: No

Reviewer #2: Yes

Reviewer #3: Yes

Reviewer #4: Yes

Reviewer #5: N/A

Reviewer #6: N/A

3. Have the authors made all data underlying the findings in their manuscript fully available?

Reviewer #1: Yes

Reviewer #2: Yes

Reviewer #3: Yes

Reviewer #4: Yes

Reviewer #5: Yes

Reviewer #6: No

4. Is the manuscript presented in an intelligible fashion and written in standard English?

Reviewer #1: Yes

Reviewer #2: Yes

Reviewer #3: No

Reviewer #4: Yes

Reviewer #5: Yes

Reviewer #6: Yes

5. Review Comments to the Author

Reviewer #1: This manuscript presents an important contribution to the field of urban studies and offers valuable insights into the historical and contemporary significance of fortification sites. The suggestions are as follows:

The term "stratification process" is an important concept but has not been clearly defined in the manuscript. It is recommend to providing a concise explanation of what this entails in the context of fortifications and urban landscapes.

The current literature review could be expanded to better situate your findings in the broader academic discourse. Discussing previous studies on urban fortifications and their transformations will demonstrate the novelty of your research and highlight existing gaps that your work addresses.

Methodology: How were the data collected? What criteria were used for the selection of case studies? It is recommend to including this information to improve the reliability of the research.

Consider exploring how specific socio-political and historical contexts in each city shaped the unique outcomes of their fortification sites. These differences could provide deeper insights into the stratification processes you describe. In the discussion part, the authors could discuss more about how can the lessons from these three cities be applied to similar projects in other cities?

Reviewer #2: The manuscript provides a comprehensive perspective to understand the similarities and differences of different cities in dealing with historical fortification sites by comparing three cities under different cultural and political backgrounds, and provides meaningful experience for the protection and development of fortification sites in other cities. However, the manuscript still needs to be improved in the following aspects:

1.The reason for dividing into three stages is not clearly explained. Why are these three periods? Judging from Figure 14, the time spans of these three fortifications are very different. Moscow's fortifications span 200 years, while Beijing's only has 50 years. Their phases are the same. What is the reason behind this?

2.The manuscript, while providing a comprehensive perspective, lacks a crucial element-a three-dimensional analysis of the fortifications of the three cities. The urban landscape, with its planar and three-dimensional changes, is a key aspect of understanding urban history. It is recommended to add an analysis of the three-dimensional aspect through photos or 3D models as this will significantly enhance the readers' understanding of the subject.

3.The discussion of strategies in each period is relatively general. To truly understand the differences in strategies among the three cities, it is essential to have specific strategy analysis diagrams. These could include analyzing the changes in city walls, roads, greenery, and pedestrian spaces from a sectional perspective. Figure 15, while informative, is also quite general and does not clearly depict the differences in strategies. Therefore, the addition of specific strategy analysis diagrams is recommended to enhance the readers' understanding.

4.Minor error: On line 491, "Comparative Analysis of the Three Cities in Phase Two", it should be Phase Three.

Reviewer #3: Many typographical errors throughout the entire document. Odd spacing, parts of words, and inconsistent capitalizations.

Although the research is solid, the main premise of the research, beautiful city, is not clearly laid out. No background, examples, or definitions are provided, only general terms about harmonious development. It is unclear if this is a strategic design agenda (a Beautiful City initiative) or just a category developed by the authors (generally designing beautiful cities). This needs to be corrected for the findings and conclusions to be accurately assessed. Currently, there is no clear description of the “beautiful city” development practices.

Additionally, tourism is described as “green, low-carbon, and environmentally friendly nature of tourism” in the last paragraph of page six. This is a broad assumption with no supporting data. Furthermore, one of the ways tourism and urban development is later assessed is the increase in car ownership. This creates a paradox - - how can tourism be “green” but it’s success be measured on the increase in car-ownership?

Lastly, the overall purpose of the research is unclear. What is the intention of this coupling model? There is no clear need or application offered.

Reviewer #4: Comments and Suggestions for Authors

Based on the perspective of HUL, this paper discusses the evolution of original sites of fortifications in three cities. The paper is well written and has novel ideas, but there are some areas where the clarity and accuracy of scientific reporting could be improved:

Abstract and Introduction: An explanation of the results can be appropriately included in the summary, so that readers can get the relevant information more quickly. The introduction may appropriately set the background of the fortifications, but additional explanations of key theories and frameworks can enhance the reader's understanding, especially for interdisciplinary readers unfamiliar with the subject.

Text description: The overall structure of the article can be more clear. For example, in the case analysis of each city, a unified analytical framework can be adopted, such as following the sequence of "historical evolution of the fortress - demolition and planning - construction process and problems - reflection and renewal strategies", so that readers can compare the similarities and differences between different cities more easily.

Methodology: This paper studies the historical landscape of the city from HUL's novel perspective, but it does not clearly point out how the method is realized. At the same time, the frequency of HUL mentioned in the whole paper is low.

Results Presentation: The overall structure of the article is rigorous, but suggestions for city builders can be added in the conclusion part by summarizing these three case cities.

Discussion and Conclusion: When discussing the similarities and differences of the evolution of the three urban fortress sites, the depth of analysis can be further deepened. For example, in the common part, in addition to pointing out similar stages and planning methods, we can also explore the deep-rooted reasons behind these commonalities, such as the macro trend of global urbanization process, and the general influence mechanism of similar urban development needs (such as traffic improvement, population accommodation, etc.) on the evolution of fortress sites.

Scientific Language: Overall, the language is formal and suitable for a scientific audience. But some parts of the paper, particularly in the literature review and conclusion, could be more concise.

Limitations: The study identified some key limitations, such as not having enough long-term, continuous data, which is important for scientific transparency. Expanding the generality of how these limitations affect the findings will strengthen the paper.

Reviewer #5: Dear authors,

Thank you very much for the opportunity to read your paper. From a general perspective your topic is a real interest one, even the research is mostly at a descriptive level. So, to increase the consistency of your paper I recommend to include some statistics about the profile of these alpha cities (population, PIB, density etc), preferably with a global perspective (the rank of each city by various criteria into a global score).

Also, please review carefully your text, I suppose that at 491 it is actually Phase Three and not again Two (as it is at 318).

Reviewer #6: The article examines the transformation of historic fortifications in Paris, Beijing, and Moscow over time. While initial plans included green and walkable spaces, actual deconstruction (of fortified area) and their redevelopment often sacrificed those plans to provide for automobiles. This led to typical urban ills including increasing pollution, decreasing connectivity and so forth. However, all three of these cities are now in the process of rethinking their approaches to these historic sites.

Overall, this article is an engaging read and presents an interesting comparison of three phased processes in three major cities of the world. The analysis of the parallels between the cities is fascinating but I was left wondering how they diverged. I think more engagement with that as well as the broader planning history literature is needed. Also, the crux of the analysis focuses on one smallish area per city, yet no mention is given to how that particular area within each city was chosen (apologies if I missed it) nor is evidence provided as to how representative each particular area is to the changes occurring along the rest of the fortification areas in the cities.

Major comments:

• Lines 1-57 seem to be lacking references. Same for lines 87-94. Same for lines 190-204. More engagement with the existing literature is needed throughout.

• I think it would be useful to further the discussion of the economic, military and political reasons for dismantling the fortifications. This could perhaps be in the form of a table with the three sites as columns, and econ, military, and political reasons as rows with bullet points for each row/sites.

• In Figures 9 and 12, I think it would be helpful to label the streets (especially Blvd des Marechaux and the Périphérique). Similarly, any street or building mentioned in the text, should ideally be labeled on the respective figures to aid in orienting the reader who might not be familiar with the layout of all three of these cities especially on such a fine scale.

• I think the paper would benefit from having a series of pictures of the modern day setting of the three areas of focus (from Google Street View perhaps). It would help situate statements such as this one “Galaxy SOHO and the Petroleum Building, are enormous, not only creating a significant sense of oppression for pedestrians but also severely clashing with the surrounding urban image”

• It is difficult to compare the figures to each other and see the evolution of the different land uses over time. Is there a way to combine these figures so that they are more easily comparable? Also, since these look like GIS files, could you calculate the area devoted to each land use for each city and graph it over time (as a new figure)? Ideally, this would be both for the three smallish areas you focus on and the entire fortified ring area.

• To synthesize your research, maybe you should talk about the process of place making around the fortifications that has taken a long time but that seems to be coming to fruition (finally – after a few setbacks), but how under the guide of the HUL the fortified areas have evolved into pedestrian/bike networks but also important from recreation, environmental sustainability issue as well as cultural stand points (although I’m not sure that was the case in all three cities- perhaps something to discuss in more detail).

• Bringing the discussion back to the HUL is needed in the conclusions.

• I don’t think you should have an abbreviation in the title.

Minor comments:

• Lines 95-104 instead of “this study conduct”, “It analyzes” and other instances, consider saying “we conducted”, “we analyzed”.

• Fig 1. Consider increasing the font on the legend (and a little on the scale bar). Also, I think it would be great to have an inset map that shows a close up (and extent rectangle) of where the sites you focus on in subsequent figures are located relative to the cities themselves, i.e. where is Porte de Vanve in Paris, etc.

• L104, the mention of Figure 1 here seems misplaced. Maybe add it early on when you justify the focus on those three cities?

• Throughout, the periods should come after the parenthetical references, for example, “(10).”

• Figures 4 and 5, since green spaces are the light green, and you are showing trees as the small darker green stars (I’m assuming although that needs to be added to the legend), it’s a bit confusing to have light green with dark green dots as No-building zones (unless you mean parks, or vegetated walkable spaces but then please clarify)?

• When you first mention Hua Nangui and Liang Sicheng, please state who they are (pardon my ignorance).

• Table 1, “construction time” is the start of the construction? Because it took more than a year (at least for Paris), no?

• L223 (but check for other instances for ex L231) the “the” is missing before “Garden ring”

• Also, L226, the verb is missing for “new bridges over the Moscow River” (probably “were built”)

• Lastly, I think that the Moscow River is actually (even in English) referred to as the Moskva River. Similarly, Boulevards of Marshals is called the Boulevard des Maréchaux. I would be inclined to suggest it might be best not to translate all these place names.

• L227, do you mean “in and around the garden rings”?

• L235 “Owing to the Garden Ring’s excessive number of lanes, some sections have up to eight lanes, encroaching on the city’s public space” this sentence needs to be rephrased.

• Somewhere around L260 I think it would be beneficial to mention that the Périphérique is just on the outside of the former fortifications (nearly adjacent to it and essentially marks the boundary between the City of Paris and the suburbs) and that the Périphérique is the reason for the difficulties in going from the City of Paris to the suburbs; not the former fortifications and roads that are within that zone.

• L312, typo “These”

• Consider combining Figures 14 and 15 into one figure as you have a lot of figures and I think it would help comprehension and legibility.

• L384-385 “a 14-kilometer noise barrier” around what?

• L386, what area exactly is “the Crown”?

• L414, add “as well as” between “continuity. Increase”

• L420, maybe specify that the V stands for Vélo (or Bicycle) and not the roman numeral five.

6. PLOS authors have the option to publish the peer review history of their article (what does this mean? ). If published, this will include your full peer review and any attached files.

**Do you want your identity to be public for this peer review?** For information about this choice, including consent withdrawal, please see our Privacy Policy .

Reviewer #1: No

Reviewer #2: No

Reviewer #3: No

Reviewer #4: No

Reviewer #5: **Yes: ** Bogdan NADOLU

Reviewer #6: No

---

## [Author Response · Author response to Decision Letter 1]

16 Jan 2025

Reviewer 1

Q：The term "stratification process" is an important concept but has not been clearly defined in the manuscript. It is recommended to providing a concise explanation of what this entails in the context of fortifications and urban landscapes.

A：Thank you for your comments.In the revised manuscript, I have provided a detailed explanation of this in sections 33-43. To highlight, the approach of urban historical landscapes emphasizes the stratification of urban heritage, where these historical layers collectively contribute to the value of urban heritage. Objectively studying different segments of urban evolution helps to respect the cultural values within various historical contexts, thereby avoiding omissions or biases in value judgments.

Q：Methodology: How were the data collected? What criteria were used for the selection of case studies? It is recommended to including this information to improve the reliability of the research.

A：I found numerous historical maps on the open-source website Old Maps Online, including maps of Moscow from 1812, 1857, and 1960, as well as historical maps of Paris from 1912, 1924, and 1975. Additionally, I have KH-9 Hexagon satellite images of Beijing from 1943 and 1996. In the revised manuscript, I have utilized these historical maps, current satellite imagery, and data from OpenStreetMap to create 12 functional distribution maps for the three cities at different stages. The case selection was based on a comparison of urban morphology across different periods, focusing on areas with significant changes.

Q：Consider exploring how specific socio-political and historical contexts in each city shaped the unique outcomes of their fortification sites. These differences could provide deeper insights into the stratification processes you describe.

A：A new section has been added to the revised manuscript, titled: "Comparative Analysis of the Original Sites of Fortifications in Three Cities." This section provides a comprehensive comparison of the unique contexts surrounding the evolution of the original sites of fortifications in each city. While the timing of each stage differs across the cities, the underlying driving factors are similar, closely tied to political and economic contexts as well as the process of urbanization.

Q：In the discussion part, the authors could discuss more about how can the lessons from these three cities be applied to similar projects in other cities?

A：In the final section, I summarized three strategies. First, emphasizing ecological continuity and sustainability. Second, enhancing spatial connectivity and social integration. Lastly, introducing diverse functions to revitalize the site and prevent the original fortification sites from becoming "lost spaces." These strategies are equally applicable to other cities facing similar cultural heritage projects.

Reviewer 2

Q：The reason for dividing into three stages is not clearly explained. Why are these three periods? Judging from Figure 14, the time spans of these three fortifications are very different. Moscow's fortifications span 200 years, while Beijing's only has 50 years. Their phases are the same. What is the reason behind this?

A：Thank you for your feedback.Although the stages occurred at different times in the three cities, the underlying driving forces are similar, closely tied to political and economic conditions as well as the process of urbanization. Firstly, Beijing entered the urbanization process relatively late, beginning after 1949, whereas European cities started much earlier. Furthermore, although Paris entered urbanization relatively early, it paradoxically constructed new fortifications in 1841 while other countries were dismantling theirs. This decision, regarded as a mistake by policymakers, can be seen as a "reverse urbanization" process. It was not until 1919, when the fortifications were proven entirely useless, that Paris dismantled them and officially entered the first stage. The remaining parts of this process are elaborated in detail in the newly added section, "Comparative Analysis of the Original Sites of Fortifications in Three Cities."

Q：The manuscript, while providing a comprehensive perspective, lacks a crucial element-a three-dimensional analysis of the fortifications of the three cities. The urban landscape, with its planar and three-dimensional changes, is a key aspect of understanding urban history. It is recommended to add an analysis of the three-dimensional aspect through photos or 3D models as this will significantly enhance the readers' understanding of the subject.

A：To enhance readability, the revised manuscript includes 12 functional distribution maps of the original fortification sites in three cities at different stages. Additionally, 9 strategy analysis diagrams in 3D models and 2 urban road sectional diagrams have been added in the sections discussing the renewal strategies for the fortification sites in each city.

Q：The discussion of strategies in each period is relatively general. To truly understand the differences in strategies among the three cities, it is essential to have specific strategy analysis diagrams. These could include analyzing the changes in city walls, roads, greenery, and pedestrian spaces from a sectional perspective. Figure 15, while informative, is also quite general and does not clearly depict the differences in strategies. Therefore, the addition of specific strategy analysis diagrams is recommended to enhance the readers' understanding.

A：For the three cities, I have supplemented the revised manuscript with specific strategies to make them more concrete, clear, and directly operable at the physical level. These strategies are illustrated through 9 3D model diagrams. In Paris, the strategies include transforming the ring road, covering expressways to create public spaces, constructing buildings on expressways, increasing green space, reshaping landscape corridors, utilizing the spaces under elevated bridges as public spaces, redefining building functions, and formulating temporary planning measures. In Beijing, the strategies involve creating urban greenways, utilizing the remnants of city walls and moats to shape public spaces, and improving riverside green spaces. For Moscow, the strategies focus on optimizing traffic layouts, widening sidewalks, and reshaping spatial nodes to enhance the original sites of the fortifications.

Reviewer 4

Q：Methodology: This paper studies the historical landscape of the city from HUL's novel perspective, but it does not clearly point out how the method is realized. At the same time, the frequency of HUL mentioned in the whole paper is low.

A：Thank you for reviewing my manuscript.Based on your suggestions, I have revised the structure of the article. Firstly, the analysis for each city is presented in the sequence of "demolition and planning — construction process and problems — reflection and renewal strategies." Subsequently, a comparative analysis of the three cities is conducted based on the quantitative results, and finally, the commonalities in the evolution of the original sites of fortifications across the three cities are summarized.

Q：Results Presentation: The overall structure of the article is rigorous, but suggestions for city builders can be added in the conclusion part by summarizing these three case cities.

A：By summarizing the renewal strategies for the original sites of fortifications in the three cities, I have distilled three key insights, which have been included in the conclusion section of the article. First, emphasizing ecological continuity and sustainability. Second, enhancing spatial connectivity and social integration. Lastly, introducing diverse functions to revitalize the site and prevent the original fortification sites from becoming "lost spaces." These strategies are equally applicable to other cities facing similar cultural heritage projects.

Q：Discussion and Conclusion: When discussing the similarities and differences of the evolution of the three urban fortress sites, the depth of analysis can be further deepened. For example, in the common part, in addition to pointing out similar stages and planning methods, we can also explore the deep-rooted reasons behind these commonalities, such as the macro trend of global urbanization process, and the general influence mechanism of similar urban development needs (such as traffic improvement, population accommodation, etc.) on the evolution of fortress sites.

A：A new section has been added to the revised manuscript, titled: "Comparative Analysis of the Original Sites of Fortifications in Three Cities." This section provides a comprehensive comparison of the unique contexts surrounding the evolution of the original sites of fortifications in each city. While the timing of each stage differs across the cities, the underlying driving factors are similar, closely tied to political and economic contexts as well as the process of urbanization.

Q：Limitations: The study identified some key limitations, such as not having enough long-term, continuous data, which is important for scientific transparency. Expanding the generality of how these limitations affect the findings will strengthen the paper.

A：In the final paragraph of the article, I have added a discussion of its limitations. First, the case studies are limited to three cities. While these cities are representative, the findings may not fully apply to cities with different cultural contexts and developmental stages. Second, the quantitative analysis relies heavily on the ArcGIS Pro platform and is constrained by the completeness and accuracy of historical data. As a result, functional changes during certain periods could only be indirectly inferred. Furthermore, the stratified historical analysis lacks depth in linking specific details to historical events, which limits its ability to fully capture the dynamic changes and their impacts across different periods.Future research should expand the scope of study and incorporate multi-dimensional datasets to enhance the dynamic analysis of the original sites of fortifications. These efforts would provide more comprehensive theoretical and practical guidance for heritage preservation and sustainable urban development.

Reviewer 5

Q：From a general perspective your topic is a real interest one, even the research is mostly at a descriptive level. So, to increase the consistency of your paper I recommend to include some statistics about the profile of these alpha cities (population, PIB, density etc), preferably with a global perspective (the rank of each city by various criteria into a global score).

A：Thank you very much for your comments; you are correct. However, I currently do not have sufficient data to provide statistics for all cities. To address the issue of the study relying too heavily on descriptive analysis, I have added a quantitative section to the paper. Using historical maps, I calculated the functional proportions of the original fortification sites in the three cities during different stages. By analyzing the patterns of change, I found that the evolution of the original fortification sites in the three cities exhibits significant commonalities, driven by underlying factors such as political and economic contexts and the pace of urbanization.

Reviewer 6

Q：I think it would be useful to further the discussion of the economic, military and political reasons for dismantling the fortifications. This could perhaps be in the form of a table with the three sites as columns, and econ, military, and political reasons as rows with bullet points for each row/site.

A：Thank you for your comments. I have added a new section to the revised manuscript titled "Comparative Analysis of the Original Sites of Fortifications in Three Cities." This section provides a comprehensive comparison of the underlying factors driving the morphological evolution of the original fortification sites in each city at different stages, including political, military, and economic elements. The results indicate that, although the stages occurred at different times across the cities, cities in the same stage show significant similarities in their political and social contexts as well as in their urbanization processes.

Q：In Figures 9 and 12, I think it would be helpful to label the streets (especially Blvd des Marechaux and the Périphérique). Similarly, any street or building mentioned in the text, should ideally be labeled on the respective figures to aid in orienting the reader who might not be familiar with the layout of all three of these cities especially on such a fine scale.

A：To help readers better understand the location of the case study areas within each city, I have added a legend for each city. The legend indicates the location of the ring roads as well as the specific location of the case study areas.

Q：I think the paper would benefit from having a series of pictures of the modern day setting of the three areas of focus (from Google Street View perhaps). It would help situate statements such as this one “Galaxy SOHO and the Petroleum Building, are enormous, not only creating a significant sense of oppression for pedestrians but also severely clashing with the surrounding urban image”

A：Due to copyright issues, I am unable to include online photos or Google Street View images in the paper, and since I am not currently in the three cities, I cannot take photos. To address the lack of photos, I have added 9 3D model illustrations to help readers better understand the strategies of each city.

Q：It is difficult to compare the figures to each other and see the evolution of the different land uses over time. Is there a way to combine these figures so that they are more easily comparable? Also, since these look like GIS files, could you calculate the area devoted to each land use for each city and graph it over time (as a new figure)? Ideally, this would be both for the three smallish areas you focus on and the entire fortified ring area.

A：Following your suggestions, I have created 12 maps in ArcGIS Pro, showing the geographic data of three cities at four different time points to facilitate the comparison of changes at each stage in the three cities. Additionally, I calculated the functional proportions of the original fortification sites in different stages and found strong commonalities in their evolution. The driving factors behind these changes are closely related to political, economic, and urbanization speed.

Q：To synthesize your research, maybe you should talk about the process of place making around the fortifications that has taken a long time but that seems to be coming to fruition (finally – after a few setbacks), but how under the guide of the HUL the fortified areas have evolved into pedestrian/bike networks but also important from recreation, environmental sustainability issue as well as cultural stand points (although I’m not sure that was the case in all three cities- perhaps something to discuss in more detail).

A：In terms of strategies, I have added more details and expressed them through 9 3D model diagrams. In Paris, the proposed strategies include: transforming the ring road, covering highways to increase public space, constructing buildings on highways, increasing green space, reshaping landscape vistas, utilizing spaces under elevated bridges for public use, rethinking building functions, and implementing temporary planning. Beijing’s strategies involve creating urban greenways, making full use of the remains of the city walls and moats for public space creation, and improving riverside green spaces. Moscow's strategies focus on optimizing traffic layouts, widening sidewalks, and reshaping spatial nodes to enhance the original fortification sites.

Q：Bringing the discussion back to the HUL is needed in the conclusions. I don’t think you should have an abbreviation in the title

A：Thank you for your reminder. I have revisited the discussion of HUL in the conclusion section and revised my title by removing the abbreviation.

---

## [Editor Report · Decision Letter 1]

21 Jan 2025

A study on the evolution of original sites of fortifications from the perspective of Historic Urban Landscape: Cases of Paris, Beijing, and Moscow

PONE-D-24-28746R1

Dear Dr. Mo Xu,

We’re pleased to inform you that your manuscript has been judged scientifically suitable for publication and will be formally accepted for publication once it meets all outstanding technical requirements.

Kind regards,

Samuel Kofi Tchum, Ph.D.

Academic Editor

PLOS ONE
---

## [Editor Report · Acceptance letter]

PONE-D-24-28746R1

PLOS ONE

Dear Dr. Xu,

I'm pleased to inform you that your manuscript has been deemed suitable for publication in PLOS ONE. Congratulations! Your manuscript is now being handed over to our production team.

Kind regards,

on behalf of

Dr Samuel Kofi Tchum

Academic Editor

PLOS ONE